

# Zero to moderate methane emissions in a densely rooted, pristine Patagonian bog - biogeochemical controls as revealed from isotopic evidence

Wiebke Münchberger[1, 2], Klaus-Holger Knorr[1], Christian Blodau[1, †], Verónica A. Pancotto[3, 4], Till Kleinebecker[2, 5]

[1]Ecohydrology and Biogeochemistry Research Group, Institute of Landscape Ecology, University of Muenster, Heisenbergstraße 2, 48149 Muenster, Germany
[2]Biodiversity and Ecosystem Research Group, Institute of Landscape Ecology, University of Muenster, Heisenbergstraße 2, 48149 Muenster, Germany
[3]Centro Austral de Investigaciones Científicas (CADIC-CONICET), B. Houssay 200, 9410 Ushuaia, Tierra del Fuego, Argentina
[4]Instituto de Ciencias Polares y Ambiente (ICPA-UNTDF), Fuegia Basket, 9410 Ushuaia, Tierra del Fuego, Argentina
[5]Institute of Landscape Ecology and Resources Management, Giessen University, Heinrich-Buff-Ring 26, 35392 Gießen, Germany
[†]deceased, July 2016

*Correspondence to*: Wiebke Münchberger (wiebke.muenchberger@uni-muenster.de), Klaus-Holger Knorr (kh.knorr@uni-muenster.de)

**Abstract.** Peatlands are significant global methane ($CH_4$) sources, but processes governing $CH_4$ dynamics have been predominantly studied on the northern hemisphere. Southern hemispheric and tropical bogs can be dominated by cushion-forming vascular plants (e.g. *Astelia pumila*, *Donatia fascicularis*). These cushion bogs are found in many (mostly southern) parts of the world but could also serve as extreme examples for densely rooted northern hemispheric bogs dominated by rushes and sedges. We report highly variable summer $CH_4$ emissions from different microforms in a Patagonian cushion bog as determined by chamber measurements. Driving biogeochemical processes were identified from pore water profiles and carbon isotopic signatures. An intensive root activity within a rhizosphere stretching over 2 m depth accompanied by molecular oxygen release created aerobic microsites in water-saturated peat leading to a thorough $CH_4$ oxidation (< 0.003 mmol $L^{-1}$ pore water $CH_4$, enriched $\delta^{13}C$-$CH_4$ by up to 10‰) and negligible emissions (0.09 ± 0.16 mmol $CH_4$ $m^{-2}$ $d^{-1}$) from *Astelia* lawns. Root activity even suppressed $CH_4$ emissions from non-rooted peat below adjacent pools (0.23 ± 0.25 mmol $CH_4$ $m^{-2}$ $d^{-1}$), in which we found similar pore water profile patterns as obtained under *Astelia* lawns. Below the rhizosphere pore water concentrations increased sharply to 0.40 ± 0.25 mmol $CH_4$ $L^{-1}$ and $CH_4$ was predominantly produced by hydrogenotrophic methanogenesis. Few *Sphagnum* lawns and – surprisingly – one lawn dominated by cushion-forming *D. fascicularis* were found to be local $CH_4$ emission hot spots with up to 1.52 ± 1.10 mmol $CH_4$ $m^{-2}$ $d^{-1}$ presumably as root density and molecular oxygen release dropped below a certain threshold. The spatial distribution of root characteristics supposedly causing such pronounced $CH_4$ emission pattern was evaluated on a conceptual level that aimed to reflect extreme examples of general scenarios in densely rooted bogs. We conclude that presence of cushion vegetation as a proxy for negligible $CH_4$ emissions



from cushion bogs needs to be interpreted with caution. Nevertheless, overall ecosystem $CH_4$ emissions at our study site were probably minute compared to bog ecosystems worldwide and widely decoupled from environmental controls due intensive root activity of e.g. *A. pumila*.

## 1 Introduction

Peatland ecosystems are long-term carbon (C) stores as organic C in plant residues degrades only incompletely and accumulates over millennia (Gorham, 1991; Yu, 2012). While being C sinks, peatlands are significant natural methane ($CH_4$) sources on the global scale responsible for about 10% of global annual CH4 emissions (Aselmann and Crutzen, 1989; Mikaloff Fletcher et al., 2004). $CH_4$ has a 28fold higher global warming potential over a 100-year time horizon compared to carbon dioxide ($CO_2$) (IPCC, 2014) and, thus, processes governing $CH_4$ production, consumption and emissions from peatlands

receive much attention to estimate global greenhouse gas emissions.

The slowdown of decomposition and accumulation of C in rainwater-fed peatlands (bogs) results from e.g. a high water table, anoxic conditions, recalcitrant peat forming litter, inactivation of oxidative enzymes and an accumulation of decomposition end-products (Beer and Blodau, 2007; Limpens et al., 2008; Bonaiuti et al., 2017). The upper, unsaturated and oxic peat layers with high decomposition rates typically extent only about few decimetres (Whalen, 2005; Limpens et al., 2008). Once

methanogenic conditions are established, $CH_4$ production is mainly controlled by substrate supply of degradable organic matter (Hornibrook et al., 1997; Whalen, 2005). Produced $CH_4$ is released from peat by three main pathways: diffusion, ebullition and plant-mediated transport (Blodau, 2002; Limpens et al., 2008). During slow diffusive transport along the concentration gradient from deep anoxic peat layers to the atmosphere, $CH_4$ might get trapped within upper, oxic zones and consumed by methanotrophic microbes (Chasar et al., 2000; Whalen, 2005; Berger et al., 2018). Consequently, water table position and

fluctuations that determine the extent of oxic zones strongly control the amount of emitted $CH_4$ (Blodau and Moore, 2003; Whalen, 2005). While $CH_4$ oxidation by methanotrophs results in suppressed $CH_4$ emissions, ebullition by fast release of gas bubbles can substantially increase $CH_4$ emissions (Fechner-Levy and Hemond, 1996; Whalen, 2005; Knoblauch et al., 2015; Burger et al., 2016). Many wetland plants establish aerenchymatic root tissues to supply roots growing in anoxic peat with molecular oxygen ($O_2$) (Joabsson et al., 1999; Colmer, 2003). Yet, these aerenchymatic roots have frequently been described

to act as conduits to the atmosphere for $CH_4$ transport bypassing oxic surface layers thereby preventing $CH_4$ oxidation and enhancing emissions (e.g. Joabsson et al., 1999 and references therein; Chasar et al., 2000; Colmer, 2003, Whalen, 2005, Knoblauch et al., 2015; Berger et al., 2018). Vegetated or open water pools that are characteristic of peatlands have received less attention than the vegetated surfaces (Pelletier et al., 2014). However, pools are considered as strong $CH_4$ emitters due to their anoxic conditions (Hamilton et al., 1994; Blodau 2002; Burger et al., 2016) that can even turn the peatlands' C balance

into a source (Pelletier et al., 2014) but examples of low emission pools have been also reported (Knoblauch et al., 2015).

Carbon isotopic signatures are a valuable tool for identifying mechanisms and pathways of $CH_4$ production and consumption. Carbon isotopic signatures in pore water of northern hemispheric bogs (hereafter termed northern bogs) vary typically between -80 to -50‰ and -25 to -0‰ for $CH_4$ and $CO_2$, respectively (e.g. Whiticar et al., 1986; Hornibrook et al., 1997; Chasar et al.,




2000; Hornibrook et al., 2000; Beer et al., 2008; Steinmann et al., 2008; Corbett et al., 2013). Depending on the available substrate for methanogenesis, it was mostly observed that either the acetoclastic or hydrogenotrophic pathway predominates, resulting in distinctive carbon isotope signatures of $CH_4$ (acetate fermentation $\delta^{13}C$-$CH_4$ ~ -65 to - 50‰, $CO_2$ reduction $\delta^{13}C$-$CH_4$ ~ -110 to -60‰) (Whiticar et al., 1986). Methane is prevailingly produced by acetoclastic methanogenesis when fresh,

labile organic matter is available whereas the production pathway shifts towards hydrogenotrophic methanogenesis when organic matter becomes increasingly recalcitrant (Hornibrook et al., 1997; Popp et al., 1999; Chasar et al., 2000; Conrad, 2005). A predominance of the latter pathway is indicated by strong fractionation between $^{12}C/^{13}C$ resulting in relatively depleted $\delta^{13}C$-$CH_4$ corresponding to a fractionation factor $a_c$ between 1.055 and 1.090 (Whiticar et al., 1986) and a $\delta^{13}C$-$CH_4$ isotopic signature following that of $\delta^{13}C$-$CO_2$ (Hornibrook et al., 2000). Subsequent methanotrophy alters the isotopic signature

of $CH_4$ by discriminating against $^{13}C$ and the residual $CH_4$ remains enriched while produced $CO_2$ becomes depleted in $^{13}C$ (Chasar et al., 2000; Popp et al., 1999). As both, acetoclastic methanogenesis and methanotrophy, result in a $\delta^{13}C$-$CH_4$ signature enriched in $^{13}C$ and a small apparent fractionation factor (Whiticar et al., 1986; Conrad, 2005), the acetoclastic pathway cannot be clearly separated from methanotrophic conditions based on carbon isotopes only (Conrad, 2005).

The majority of studies elucidating mechanisms governing $CH_4$ production, consumption and emission have been conducted

in peatlands on the northern hemisphere (e.g. Blodau, 2002; Limpens et al., 2008; Yu, 2012 and references in these), whereas only little research deals with $CH_4$ emissions and their underlying controls in bogs on the southern hemisphere (hereafter termed southern bogs) (Broder et al., 2015). Contrary to those northern bogs, southern bogs are not only dominated by *Sphagnum* mosses, dwarf shrubs or few graminoids such as the rush-like species *Tetroncium magellanicum*, but also by vascular plants (Kleinebecker et al., 2007) that can densely root upper peat layers (Fritz et al., 2011, Knorr et al., 2015). These

vascular plants, for instance of the genus *Astelia* (Asteliaceae), *Donatia* (Stylidiaceae) or *Oreobolus* (Cyperaceae), independently developed a cushion life form to protect from harsh climate in cold environments (Gibson and Kirkpatrick, 1985; Boucher et al., 2016). Bogs dominated by cushion-forming vascular plants (hereafter termed cushion bogs) can be found in many (mostly southern) parts of the world, for instance all along the high Andes (Coombes and Ramsay, 2001; Benavides et al., 2013; Fonkén, 2014) down to southernmost Patagonia (Ruthsatz and Villagran, 1991; Heusser, 1995; Kleinebecker et

al., 2007; Grootjans et al., 2014), in highlands of eastern Africa (Dullo et al., 2017) as well as in New Guinea (Hope, 2014), New Zealand and Tasmania (Gibson and Kirkpatrick, 1985; Ruthsatz and Villagran, 1991) and some sub-Antarctic islands (Ruthsatz and Villagran, 1991). These cushion bogs could also be regarded as extreme examples for densely rooted northern bogs that are dominated by rushes and sedges.

To our knowledge, only two studies reported $CH_4$ emissions from southern vascular-plant dominated bogs with quite

inconsistent results. Over a New Zealand bog dominated by the evergreen "wire rush" *Empodisma robustum*, $CH_4$ emissions exceeded those commonly reported from northern bogs (Goodrich et al., 2015), while emissions from a Patagonian cushion bog dominated by *Astelia pumila* were negligible (Fritz et al., 2011). Both, high and negligible $CH_4$ emissions, were mainly explained by the extensive aerenchymous roots of the prevailing plant species: The aerenchyma might function as a conduit for $CH_4$ from deep peat layers to the atmosphere in the first case, whereas in the second case pronounced $O_2$ supply by roots





might have resulted in a thorough oxidation of pore water $CH_4$. Despite the variety of bogs dominated by densely rooted, aerenchymous plants on the northern and southern hemisphere, there is some evidence from latest research that very low $CH_4$ pore water concentrations could be a more common phenomenon in those ecosystems (Dullo et al., 2017; Agethen et al., 2018; Knorr et al., 2015). In general, promotion or suppression of $CH_4$ production should be determined by the ratio of root density

and activity associated with $O_2$ release versus presence of labile root organic matter or exudates accompanied with $O_2$ consumption (Blodau, 2002; Agethen et al., 2018).

Fritz et al. (2011) were the first to investigate the $CH_4$ biogeochemistry below *Astelia* lawns of a Patagonian cushion bog and found emissions to be decoupled from the water table position because of considerable $O_2$ release by roots. This resulted in enhanced organic matter decomposition, oxidation of almost all $CH_4$ produced in deep peat layers and, thus, near-zero

emissions. However, $CH_4$ emissions were determined only sporadically and not all predominant microforms and plant communities were considered, restricting general conclusions about $CH_4$ emissions from this type of ecosystem. Furthermore, more knowledge on biogeochemical processes throughout the peat column of *Astelia* lawns and other microforms is needed to extent our limited understanding of these first insights into $CH_4$ dynamics in cushion bogs.

Here, we quantified austral summer $CH_4$ emissions from dominant microforms in a Patagonian cushion bog and examined

possible environmental and biogeochemical controls. We hypothesized that (I) emissions from cushion plant vegetation dominated by *Astelia pumila* or *Donatia fascicularis* are negligible while pools and *Sphagnum* lawns emit $CH_4$ in considerable amounts, and that (II) carbon isotopic signatures obtained from pore water below *Astelia* lawns reflect a distinct $CH_4$ oxidation effect whereas pore water composition in non-rooted peat below pools remains less affected, and that (III) besides by methanotrophy, isotopic composition of pore water $CH_4$ could also be affected by a predominance of acetate fermentation

throughout the rhizosphere of *Astelia* lawns shifting towards $CO_2$ reduction with peat depth and in non-rooted peat below pools. Biogeochemical controls of $CH_4$ emissions from *Sphagnum* lawns were not subject of this study, as they have been intensively studied in northern bogs (e.g. Hornibrook et al., 1997; Beer and Blodau, 2007; Steinmann et al., 2008; Corbett et al., 2013) and even southern bogs with mixed vegetation in absence of notable cushion plant coverage (Broder et al., 2015).

## 2 Material and Methods

### 2.1 Description of the study site

The study was conducted at a cushion bog on the Peninsula Mitre in southernmost Patagonia, Tierra del Fuego (Moat, 54°58'S, 66°44'W, Fig. 1). Peatlands of the Peninsula Mitre provide the nowadays rare opportunity to study carbon dynamics under pristine conditions as they can be considered as largely undisturbed by human activities such as drainage, agriculture or elevated atmospheric nitrogen deposition (Kleinebecker et al., 2008; Fritz et al., 2011; Grootjans et al., 2014; Paredes et al.,

2014). About 45% of this area are covered by peatlands (Iturraspe, 2012) while little is known about these ecosystems because of their poor accessibility (Iturraspe, 2012). The study site at Moat belongs to a complex system of sloping mires, blanket bogs, fens and – in coastal areas - cushion bogs (Iturraspe, 2012; Borromei et al., 2014). The studied cushion bog is located in



exposed proximity to the Beagle channel with harsh winds (Grootjans et al., 2014) resulting in an oceanic climate with average daily temperatures of 6°C and annual precipitation of 500 mm (Fritz, 2012).

Large areas of the cushion bogs in Moat are covered by the cushion-forming plants *Astelia pumila* and *Donatia fascicularis*. The vegetation of the studied bog was composed of a mosaic of different plant communities characterized by specific plant species with lawns of *A. pumila* being the prevailing microform. Sparsely vegetated pools of different size (< 0.5 m² - ~10 m²) and ~0.5 m depth were embedded in these *Astelia* lawns. *Sphagnum magellanicum* or *D. fascicularis* grew in small lawns (patches of few square meters) in close vicinity to the pools which we assume to provide protection against desiccation for *S. magellanicum*. *Tetroncium magellanicum*, a rush-like herb that does not form cushions but presumably establishes aerenchymous roots as it tolerates inundation (von Mehring, 2013), was associated predominantly with *Sphagnum* lawns but also with *Astelia* or *Donatia* lawns. Peat formation started ~11000 years ago (Borromei et al., 2014), and previously, the bog was dominated by *S. magellanicum* while *A. pumila* invaded the area around 2600 years before present as determined by pollen analyses (Heusser, 1995). The peat below *Astelia* lawns was densely rooted with on the average 2.15 g DW L$^{-1}$ down to 1.7 m (Fritz et al., 2011). Mean root lifetimes of *A. pumila* has been estimated to be ~3-4 years indicating a high root turnover (Knorr et al., 2015).

## 2.2 Sampling and analysis of solid peat and root biomass

Peat coring in *Astelia* and *Sphagnum* lawns (data not shown) was done using a Russian peat corer (Eijkelkamp Agrisearch Equipment, Giesbeek, the Netherlands) down to a maximum depth of 7.5 m in *Astelia* lawns. Roots of *A. pumila* in *Astelia* lawn cores were sorted out and treated separately. The density of *D. fascicularis* root biomass was determined by cutting three 0.1 x 0.1 m sods down to a depth of 0.4 m. Peat and root biomass samples were oven dried at 70°C until constant weight to calculate peat bulk density and porosity as well as *D. fascicularis* root density. Total C and N contents together with abundance of $^{15}$N and $^{13}$C in the solid peat were determined using an elemental analyser (EA 3000, EuroVector, Redavalle, Italy) connected to an isotope ratio mass spectrometer (IRMS, NU instruments, Wrexham, UK).

## 2.3 Environmental variables

Environmental variables were determined during two measurement campaigns in 2015 and 2016 at half-hourly intervals. Photosynthetic active radiation (PAR, HOBO S-LIA-M003, Onset, USA, up to 2500 µmol m$^{-2}$ s$^{-1}$) and air temperature (HOBO S-TMB-M0x, Onset, USA) were recorded by a weather station (HOBO U30 NRC, Onset, USA) in both years during austral summer months. Water table fluctuations at two replicate sites in *Astelia* lawns (Levelogger Edge 3001 and Barologger Edge, Solinst, Canada; both installed in perforated PVC tubes) and soil temperature in four depths of 0.05, 0.1, 0.3 and 0.5 m (HOBO TMCx-HD and HOBO U-12-008, Onset, USA) were measured continuously.



## 2.4 Chamber measurements and analyses of soil-atmosphere CH₄ fluxes

### 2.4.1 Field measurements

The closed chamber technique was used to determine $CH_4$ fluxes during two measurement campaigns in austral summer from December 2014 to March 2015 and in February and March 2016. One week prior to measurements, PVC collars with a height

of 0.2 m were permanently installed ~0.15 m into the peat of the dominant microforms characterized by a specific plant species and a particular plant community (*Astelia* lawns: 2015 N = 3, 2016 N = 5; *Sphagnum* lawns: 2015 N = 3, 2016 N = 5; *Donatia* lawns: 2016 N = 3). The exact installation depth and the micro-relief of the surface within each collar was determined repeatedly to calculate the exact headspace volume for $CH_4$ flux estimation. The collars were equally arranged around three wooden platforms constructed in January 2014 to minimize disturbance during measurements and distributed over the study

site to account for spatial variability. Different numbers of replicates between the two campaigns and the different microforms are the result of logistical constraints.

A cylindrical, transparent chamber with a basal area of 0.13 $m^2$ and a height of 0.4 m was used for soil-atmosphere flux measurements. The chamber was equipped with a fan to ensure mixing of the headspace air, a PAR sensor (HOBO S-LIA-M003, Onset, USA) and a temperature sensor (HOBO S-TMB-M0x, Onset, USA). A second temperature sensor recorded

ambient temperature (HOBO S-TMB-M0x, Onset, USA) and all sensor data were logged in 1s-intervals to a data logger (HOBO U30 NRC, Onset, USA). This set-up allowed us to control air temperature inside the chamber within approximately 3°C deviation of the ambient air temperature. If necessary, ice packs were installed inside the chamber to avoid temperature increase. Due to strong wind conditions we decided against the installation of a vent tube. An opening in the top of the chamber avoided overpressure during chamber placement. This opening was closed immediately after the chamber was gently placed

on a collar for at least 3 minutes to conduct measurements. Collars were equipped with a water-filled rim to ensure a gas-tight seal between chamber and collar during measurements. Pool fluxes were performed with a floating chamber of identical dimension and design on four pools located at each platform. The chamber wall extended approximately 0.04 m into the water during measurements.

All measurements were performed under a broad range of irradiance between 7:00 am to 22:00 pm (local time) to determine

diurnal variations in $CH_4$ fluxes. At least once during each sampling period of two to three consecutive days, all selected microforms and pools were measured under dark conditions. Opaque, reflective material was used to cover the chamber for dark conditions. Measurements were taken every two weeks during the first campaign in 2015 (N = 405 measurements) and during two occasions in 2016 (N = 132 measurements). A more regular measurement routine throughout the whole season was constrained by harsh weather conditions and the remoteness of our study site.

The chamber was connected by a 2 mm inner diameter polyethylene tubing to a gas analyser (Los Gatos Ultraportable Greenhouse Gas Analyser 915-001, Los Gatos Research) to record the increase of $CH_4$ concentrations over time at a rate of 1 hz. Between measurements, atmospheric background concentrations were achieved inside the chamber. The gas analyser was



equipped with an external pump providing a flow rate of 2 L min$^{-1}$ and pumping the analysed gas back to chamber creating a closed system. Prior to each campaign, the instrument was recalibrated.

We estimated the cover of plant species within each collar to the nearest 5% and counted the number of *T. magellanicum* shoots to further characterize the different microforms (Table 1). In 2015, vegetation surveys were conducted at the beginning

and at the end of the measurement campaign. Since no change in vegetation was observed, surveys were conducted only once during the campaign in 2016. For each collar, the position above the water table was determined once during each sampling period of consecutive days to account for different positions in the micro-relief and to determine a specific water table position. Additionally, we determined the area of each individual microform patch in which a collar had been installed.

### 2.4.2 CH$_4$ flux calculation and statistical analyses

Soil-atmosphere CH$_4$ fluxes were calculated from the gas concentration increase over time within the chamber using the software package (MATLAB Release R2015a) routine described in Eckhardt and Kutzbach (2016) and Kutzbach et al. (2007). Gas concentration was modelled either as a linear or an exponential function of time. Models performance was compared using Akaike's Information Criterion (AIC) as a measure of goodness-of-fit. This routine resulted in 90% of cases in a linear function of time (N = 485) and in 10% of cases in an exponential function of time (N = 52).

We visually inspected all concentration increases over time and none of the measurements showed a stepwise concentration increase indicative of ebullient events. Of the 3 minutes measurement time, 50 - 180 s (mostly around 90 s) were selected for the CH$_4$ flux calculation to exclude unstable conditions due to e.g. chamber placement particularly at the beginning of the measurement. Fluxes with a slope not significantly different from 0 (tested with an F-Test, $p > 0.05$) were set to zero (20% of cases, N = 105).

Normal distributions of CH$_4$ flux datasets were checked by a Kolmogorov-Smirnov test. Due to non-normality, the Kruskal-Wallis analysis of variance followed by multiple comparison U-tests adjusted by the Bonferroni method were applied to determine significant differences between CH$_4$ fluxes of different microforms and collars. The relationships between CH$_4$ fluxes and environmental variables (soil temperature, water table depth fluctuations) as well as additional features of individual collars were evaluated by Spearman's rank correlation (Table 2). A response of CH$_4$ emissions to the investigated

environmental variables was identified only for a few individual collars. Therefore, no seasonal or annual ecosystem emissions were calculated. Statistical analyses were performed using Matlab (MATLAB Statistics Toolbox Release R2017a). All flux data were reported as mean ± standard deviation.

### 2.5 Depth profiles of peat pore water concentrations

### 2.5.1 Sampling and field measurements

Multilevel piezometers (MLP), as described in Beer and Blodau (2007), were installed during both measurement campaigns in austral summer 2015 and 2016 to determine pore water concentrations in depth profiles. The MLPs provided a spatial





sampling resolution of either 0.1 or 0.2 m and were equipped with diffusive equilibration samplers made of permeable silicon tubes providing 4 mL gas sample volume together with a 5 mL crimp-vial filled with deionized water and covered with a permeable membrane (Supor-200 0.2 µ, Pall corporation, Pall life sciences). To collect gas samples, MLPs were stepwise retrieved, the gas volume extracted from the equilibration sampler with a 3 mL syringe, transferred into nitrogen-flushed 5 mL

crimp vials capped with a butyl stopper with an aluminium crimp seal. MLPs were installed in *Astelia* lawns and pools in three replicates at each platform where closed chamber measurements were performed.

In January 2015, MLPs were installed in *Astelia* lawns (0.2 m resolution, 2 m depth) and pools (0.1 m resolution 0-1 m, 0.2 m resolution below down to a maximum depth of 4 m) for five weeks equilibration time. From gas samples pore water concentrations of $CH_4$, dissolved inorganic carbon (DIC) and hydrogen ($H_2$) were determined. To measure sulfate

concentrations, a potential electron acceptor originating from sea spray and affecting competitiveness of methanogenesis (Broder et al., 2015), a pore water subsample obtained from each crimp-vial was transferred to a storage vial and stored frozen until analysis. An aliquot of each remaining pore water sample was measured in-situ with a pH/EC-meter (Combo HI 98129/130, Hannah instruments, Germany).

In January 2016, MLPs were installed to a depth of 3 m in pools (resolution as in 2015) for three weeks equilibration time and

afterwards again in *Astelia* lawns (0.1 m resolution in transition zone from rooted to non-rooted peat in 1-2 m depth) for nine weeks of equilibration time. In 2016, equilibration samplers were used to collect $CH_4$ and $CO_2$ samples and determine their carbon stable isotopic signatures. Pore water samples obtained from crimp-vials were used to measure $O_2$ concentrations in-situ with a planar trace oxygen minisensor (Fibox 3, Presens, Germany).

### 2.5.2 Analytical procedures

Gas samples were transported to Germany and analysed within four weeks. Gaseous $CH_4$ and $CO_2$ concentrations were measured with a gas chromatograph (8610C, SRI Instruments, USA) equipped with a methaniser and flame ionization detector (FID). Hydrogen concentrations were analysed on a $H_2$-Analyzer (Ametek TA 3000 $H_2$ Analyser, Trace Analytical TA 3000r). Sulfate concentrations were obtained by ion chromatography (883 Basic IC plus, Metrohm, Herisau, Switzerland).

Stable carbon isotopic signatures of $CH_4$ and $CO_2$ were simultaneously determined by Cavity Ringdown Spectroscopy (CRDS;

Picarro G2201-i, Picarro Inc., USA). The instrument was calibrated in the beginning of every measurement day using two working standards of $CH_4$ (1000 ppm, -42.48‰) and $CO_2$ (1000 ppm, -31.07‰). Isotopic signatures are given in δ-notation relative to Vienna Peedee belemnite (VPDB). As samples were stored for several weeks and $\delta^{13}C$-$CO_2$ values were biased in case of high $CH_4$ concentrations in the sample, a correction procedure was applied following Berger et al. (2018).

### 2.5.3 Calculations and statistical analyses

Pore water concentrations of $CH_4$, dissolved organic carbon (DIC) and $H_2$ were re-calculated from gaseous concentrations obtained from equilibration samplers by applying Henry's law following Eq. (1):



$$c = K_H * p \tag{1}$$

where c is the concentration in mol $L^{-1}$, p the pressure in atm and $K_H$ the Henry-constant that was corrected according to Sander (1999) for mean soil temperatures of 10°C in February 2015 and 2016 as well as 7°C in April at a depth of 0.5 m. DIC concentrations at prevalent pH conditions were calculated with respect to carbonate speciation considering equilibrium constants following Stumm and Morgan (1996). All pore water data were reported as mean ± standard deviation (N = 3).

Zones of $CH_4$ production and consumption in the peat column were visually identified based on observed pore water concentration gradients. For a quantitative evaluation of pore water concentration gradients, steady-state conditions as well as a dominance of diffusive gas transport would have to be assumed; this may only partly apply to the system under study here, given the large root biomass. Nevertheless, we additionally applied a software routine using inverse modelling (PROFILE, Berg et al., 1998) for zone identification. This modelling approach supported the results obtained visually, but provided further rough estimates for production and consumption zones due to the complex diffusivity in the rhizosphere of highly rooted peat and, thus, results are presented in the appendix only.

An apparent isotopic fractionation factor $a_c$ was determined (Whiticar et al., 1986; Hornibrook et al., 2000) to assess the predominant methanogenic pathway and / or methanotrophic activity following Eq. (2):

$$a_c = \frac{\delta^{13}C_{CO_2} + 1000}{\delta^{13}C_{CH_4} + 1000} \tag{2}$$

## 3 Results

### 3.1 Characteristics of solid peat and root biomass

Roots of *A. pumila* were present in the upper profile down to a depth of 1.8 m (rhizosphere) in *Astelia* lawn cores. Throughout the rhizosphere, we observed a highly decomposed and amorphous peat supposedly originating from *A. pumila* that developed above *Sphagnum* peat. Peat below *Sphagnum* lawns was continuously formed predominantly by *Sphagnum*. Total C and N contents in the peat together with natural abundance of $^{15}N$ and $^{13}C$ are given in Table 3. Root density of *D. fascicularis* was 0.014 ± 0.010 g DW $L^{-1}$ at 0.35 m depth while root density of *A. pumila* was not determined once again as it has been already described in the literature for our study site.

### 3.2 Environmental conditions and potential controls on $CH_4$ fluxes

The study site was characterized by daily mean PAR values up to ~700 µmol $m^{-2}$ $s^{-1}$ with maximum values exceeding 2000 µmol $m^{-2}$ $s^{-1}$ and daily mean air temperatures ranging from -0.5°C to 17°C during the measurement campaigns in austral summer from January to April (Fig. 2). Soil temperature in summer reached maximum daily average values of 15°C and 12°C at 0.05 m depth and 0.5 m depth, respectively. Concurrently, the water table fluctuated close to the surface between -0.03 m and -0.23 m and differed consistently about few centimetres between the two water table measurement sites in *Astelia* lawns.



$CH_4$ fluxes were well detectable, but approached zero in many cases. For collars exhibiting low magnitude fluxes neither any seasonal trend (data not shown) nor a relationship to water table and soil temperature fluctuations could be identified (Table 2). Yet, significant relationships between water table fluctuations or variations in soil temperatures were established for those few lawns with considerable emissions (Table 2).

**3.3 Soil-atmosphere $CH_4$ fluxes and features of microforms possibly affecting $CH_4$ emissions**

Summer $CH_4$ emissions in a South Patagonian cushion bog were highly variable among the dominant microforms *Astelia* lawns, *Sphagnum* lawns and *Donatia* lawns characterized by specific plant species as well as pools and showed a pronounced spatial pattern even within microforms. Compared to all other microforms, only *Sphagnum* lawns showed considerable emissions with $1.52 \pm 1.10$ mmol $CH_4$ $m^{-2}$ $d^{-1}$ while *Astelia* lawns emitted nearly no $CH_4$ with $0.09 \pm 0.16$ mmol $m^{-2}$ $d^{-1}$ (Fig. 3). Methane emissions from pools were also low ($0.23 \pm 0.25$ mmol $m^{-2}$ $d^{-1}$) and not significantly different from intermediate *Donatia* lawn fluxes ($0.66 \pm 0.96$ mmol $CH_4$ $m^{-2}$ $d^{-1}$). During some occasions, we determined even significant, negative $CH_4$ fluxes, mainly from *Astelia* lawns (Fig. 3a, Table 2). Most of these negative fluxes as well as the majority of fluxes (41%) that were set to zero were obtained from one individual collar throughout the whole measurement campaigns precluding the possibility of a measurement artefact.

As indicated by high standard deviations, $CH_4$ emissions between replicates of all microforms were not consistent. Field observations suggested that $CH_4$ emissions of a microform were not only controlled by the predominant plant species, but might have been associated with additional features of the respective microform. Therefore additional microform features (i.e. number of *T. magellanicum* shoots, cover of *D. fascicularis*, extend of the microform in which a specific collar was installed) were assessed. However, as we did not expect such small-scale spatial variability between emissions of individual collars, the results of this survey given in Table 2 are of rather explorative, preliminary character as the low number of replicates (3 to 5 collars) in combination with several outliers barely allows any sound statistical analysis. Nevertheless, collars with elevated or comparatively high emissions had some features in common: Emissions from one *Astelia* lawn were with ~0.3 mmol $CH_4$ $m^{-2}$ $d^{-1}$ (significantly) higher in both sampling years compared to fluxes measured from the other *Astelia* lawns. This *Astelia* lawn collar with elevated emissions was characterized by (I) the presence of many *T. magellanicum* shoots together with a high share of *D. fascicularis* and (II) placed in a lawn with small extent surrounded by a mosaic of small pools and *D. fascicularis* patches (both smaller than 1 $m^2$). Surprisingly, also one *Donatia* lawn was a substantial $CH_4$ source of $2.10 \pm 0.14$ mmol $m^{-2}$ $d^{-1}$, even exceeding the highest emissions observed from *Sphagnum* lawns on all measurement occasions in 2016. Similar to the *Astelia* lawn with elevated emissions, the collars of *Donatia* and *Sphagnum* lawns with high emissions were characterized by (I) a high amount of *T. magellanicum* shoots and (II) placed in lawns surrounded by small pools and other patches of *D. fascicularis* with no *A. pumila* nearby. Emissions obtained from one *Sphagnum* lawn were with less than 0.3 mmol $CH_4$ $m^{-2}$ $d^{-1}$ in both sampling years significantly lower compared to emissions from other *Sphagnum* lawns. Vice versa,





this collar with low *Sphagnum* emissions was characterized by (I) the lowest number of *T. magellanicum* shoots among all *Sphagnum* lawn collars and (II) installed in a small *Sphagnum* lawn surrounded by *A. pumila*.

### 3.4 Pore water $CH_4$ and DIC concentration profiles

Pore water profiles in peat columns below *Astelia* lawns and pools showed similar trends in pore water concentrations and during both sampling years. $CH_4$ concentrations were almost zero in the upper pore water profile below *Astelia* lawns ($< 0.003$ mmol $L^{-1}$) down to a depth of around 1.5 m, which corresponds to the zone where the rhizosphere was most pronounced (Fig. 4a). Below this depth, $CH_4$ concentrations sharply increased up to $0.25 \pm 0.08$ mmol $L^{-1}$ at 3 m depth. $CH_4$ concentrations in pore water profiles below pools resembled profiles obtained under *Astelia* lawns on elevated levels. Throughout the upper profile down to 1.5 m, $CH_4$ concentrations were ~0.03 mmol $L^{-1}$ with a peak around 0.3 m depth ($0.06 \pm 0.06$ mmol $L^{-1}$) and increased steeply to $0.40 \pm 0.25$ mmol $L^{-1}$ in 3 m depth. Maximum concentrations in comparable depths reached similar levels in both sampling years below *Astelia* lawns, but maximum concentrations below pools were about 3times higher in 2015 compared to 2016.

DIC predominantly occurred as dissolved $CO_2$ because of the low pH ranging from 3.36 to 4.77. In contrast to $CH_4$, DIC concentrations increased constantly with depth from around 1 mmol $L^{-1}$ near the surface to $2.60 \pm 1.0$ mmol $L^{-1}$ in 2 m depth below *Astelia* lawns and $2.94 \pm 1.1$ mmol $L^{-1}$ in 3 m depth below pools (Fig. 4b). With depth, DIC converged with $CH_4$ concentrations, which was reflected by DIC:$CH_4$ ratios that were extremely high in the rhizosphere below *Astelia* lawns exceeding 100 (Fig. 4c). Beneath the rhizosphere, ratios steeply approached values below 40. Under pools, ratios slightly decreased with depth but were mostly around 40 down to 1.5 m with two distinct peaks of very little $CH_4$ at the surface and around 0.8 m depth. In deep peat layers below the rhizosphere, DIC:$CH_4$ ratios detected under *Astelia* lawns and pools converged and reached lowest ratios of ~10 and ~5, respectively.

Visual inspection of concentration profiles but also modelling of $CH_4$ production and consumption rates indicated a predominance of $CH_4$ consumption throughout the whole rhizosphere below *Astelia* lawns and in roughly corresponding depths under pools (Fig. 4a, Fig. 6). Maximum $CH_4$ consumption was identified in the lower rhizosphere around 1.5-2 m where the increase in $CH_4$ concentration was most pronounced. Deep peat layers below the rhizosphere of *Astelia* lawns and similar depths below pools were considered as $CH_4$ sources. In these depths, the high concentration levels of $CH_4$ (up to $0.40 \pm 0.25$ mmol $L^{-1}$) sustain substantial upward diffusion of $CH_4$ following the concentration gradient into the consumption zone within the rhizosphere.

### 3.5 Composition of pore water in carbon isotopic values and apparent fractionation

Values of $\delta^{13}C$ in $CH_4$ did not show a clear depth trend below *Astelia* lawns and scattered in the upper profile down to 2 m depth between $-87.2 \pm 10.1$ to $-72.1 \pm 10.3$‰ (Fig. 4d). Isotopic $\delta^{13}C$-$CH_4$ values below pools became less negative with depth





from -93.4 ± 9.2 to -73.7 ± 1.2‰ with a minimum peak in upper peat layers around 0.3 m of -101.0 ± 5.6‰. In the upper meter of the profile, values of $\delta^{13}C$-$CH_4$ were thus up to 25‰ more negative below pools compared to *Astelia* lawns while the isotopic values of both profiles converged in deep peat layers below 2 m at -80‰. Values of $\delta^{13}C$ in $CO_2$ below *Astelia* lawns became more negative with depth in the upper profile down to 1 m reaching values as low as -32.4 ± 0.8‰ and increased to -19.5 ±

1.0‰ in 3 m depth. Below pools, values became less negative with depth throughout the whole profile from -21.2 ± 0.4 to -6.0 ± 1.5‰ (Fig. 4e).

Apparent fraction factors varied only slightly and showed opposite trends below the two microforms. Values increased slightly from 1.060 to 1.066 below *Astelia* lawns, peaking at 1.045 in 0.5 m depth. Below pools, apparent fractionation factors decreased with depth from 1.079 to 1.073 with a maximum of 1.091 around 0.3 m depth (Fig. 4f).

**3.6 Hydrogen, oxygen and sulfate concentrations in pore water profiles**

Hydrogen concentrations were mostly below 3 nmol L$^{-1}$ below *Astelia* lawns while they were elevated below pools, ranging between 3 and 10 nmol L$^{-1}$ (Fig. 5a). Maximum concentrations were reached in the upper profile of *Astelia* lawns with up to 11.83 ± 17.88 nmol L$^{-1}$ and, besides a peak at the surface, below pools from 1 to 2 m depth with up to 41.61 ± 64.63 nmol L$^{-1}$. Molecular oxygen concentrations were mostly below 5% saturation and tended to be higher in the upper profile down to 2

m below *Astelia* lawns compared to pools (Fig. 5b). Sulfate concentrations were similar below both, *Astelia* lawns and pools, and reached highest values near the surface with up to 16.43 ± 11.85 µmol L$^{-1}$. With depth, concentrations decreased and approached zero below 1 m (Fig. 5c).

**4 Discussion**

This study is among the first dealing with $CH_4$ production, consumption and emissions in an austral, vascular-plant dominated

cushion bog. We aimed to reveal patterns of $CH_4$ emissions and their environmental as well as potential belowground biogeochemical controls. We furthermore attempted to elucidate the $CH_4$ dynamics in the peat below the four predominant microforms on a conceptual level based on the results presented here and previous studies.

**4.1 Environmental controls on $CH_4$ emissions**

Summer $CH_4$ emissions of a South Patagonian cushion bog dominated by *A. pumila* were low albeit spatial heterogeneity was

pronounced with small patches of *Sphagnum* or *Donatia* lawns being local emission hot spots. The variation of $CH_4$ fluxes was mainly controlled by the dominant plant species of each lawn microform which emitted significantly different amounts of $CH_4$. We found only few significant responses of $CH_4$ fluxes to water table fluctuations or variations in soil temperatures (Table 2). Such weak coupling between environmental variables and $CH_4$ emissions is not surprising as fluxes from most individual collars were relatively low. Furthermore, the water table fluctuated only slightly near the surface (Fig. 2) and may





thus not serve as a primary control on $CH_4$ emissions. Stronger effects may only be expected when amplitude and duration of fluctuations affect peat redox conditions substantially (Blodau and Moore, 2003; Knorr et al., 2009). Under such stable moisture conditions and as long as methanogenesis is not limited by substrate supply, the sensitivity of methanogenic microbial consortia to temperature should become apparent (Whalen, 2005). Variations in landscape $CH_4$ fluxes have indeed been related

to soil temperature previously (Rinne et al., 2007; Jackowicz-Korczyński et al., 2010; Goodrich et al., 2015) which is in accordance with the present study. The temporal variability of $CH_4$ emissions over the two investigated seasons was rather low (data not shown) due to low seasonality in both, temperature and precipitation, in the study region. Measurement campaigns extending over shoulder seasons and comparing years of contrasting weather conditions would be necessary to reveal the impact of more pronounced water table fluctuations or temperature regimes on $CH_4$ fluxes in South Patagonian

cushion bogs.

### 4.2 *Astelia* lawns – zero emission scenario

Emissions from *Astelia* lawns were minute ($0.09 \pm 0.16$ mmol m$^{-2}$ d$^{-1}$) and significantly lower than from all other microforms (Fig. 3) verifying our first hypothesis. The near-zero emissions were well explained by near-zero $CH_4$ concentrations ($< 0.003$ mmol L$^{-1}$) in the rhizosphere down to around 1.8 m. Only below, concentrations increased sharply to 0.2 mmol L$^{-1}$ in 2 m

depth (Fig. 4a). The low $CH_4$ concentrations even turned some *Astelia* lawns into a small sink for atmospheric $CH_4$ as shown by negative fluxes that were obtained sporadically throughout the whole measurement campaigns, confirming earlier results (Kip et al., 2012). These findings agree with previous research by Fritz et al. (2011) who observed $CH_4$ emissions from *Astelia* lawns in a comparably low magnitude as presented here and presented a similar $CH_4$ concentration depth profile. Extremely low $CH_4$ concentrations have also been obtained by Dullo et al. (2017) in upper peat layers of an Ethiopian cushion bog.

Nevertheless, in deep peat layers below the rhizosphere, $CH_4$ concentrations of our study approached magnitudes reported for Chilean bogs with mixed vegetation consisting of *Sphagnum* mosses and *A. pumila* (Broder et al., 2015). This is in the lower range of concentrations described from northern bogs mostly around 0.5 mmol L$^{-1}$ (e.g. Blodau and Moore, 2003; Beer and Blodau, 2007; Beer et al., 2008; Corbett et al., 2013) but even reaching up to 5 mmol L$^{-1}$ in comparable depths (Steinmann et al., 2008).

A $CH_4$ concentration profile as observed at our site yet suggests that in deep peat layers below the rhizosphere ($> 2$ m) $CH_4$ was produced at substantial rates and comparably low oxidation, as indicated by DIC:$CH_4$ ratios notably smaller than 40 (Fig. 4c). Subsequently, $CH_4$ got almost completely oxidized on its diffusive way upward to the atmosphere as visualized on a conceptual level (Fig. 6). Ratios of DIC:$CH_4$ under methanogenic conditions in northern bogs are typically around 5 (e.g. Hornibrook et al., 1997; Blodau and Moore, 2003; Corbett et al., 2013), but can be as high as 30 (Steinmann et al., 2008).

Notably higher ratios within the rhizosphere suggest methanotrophic activity and availability of alternative electron acceptors such as $O_2$ from aerenchymous roots suppressing methanogenesis (Colmer, 2003; Mainiero and Kazda, 2005; Knorr et al., 2008b, Fritz et al., 2011; Dullo et al., 2017). Diffusion and plant-mediated transport as pathways for upward $CH_4$ transport



would have caused higher emissions and ebullition requires supersaturation of at least ~350 mmol $CH_4$ $L^{-1}$ in the pore water (Fechner-Levy and Hemond, 1996; Beer et al., 2008). The sharp drop in $CH_4$ concentrations from -2 to -1.5 m depth characterized this peat layer as a zone of strongest methanotrophic activity, as supported by modelled consumption zones.

The assumption that root $O_2$ release by *A. pumila* and thus locally associated $CH_4$ oxidation was the most likely $CH_4$ sink was
further supported by carbon isotopic values obtained from pore water profiles that reflected rhizosphere processes. Methane isotopic signatures were close to -80‰ in deep peat layers while above 2 m depth values scattered between -90 to -70‰ throughout the rhizosphere (Fig. 4d). Methane more depleted in $^{13}C$ within the rhizosphere is likely explained by more reduced, methanogenic microsites in absence of living roots. However, less negative values revealed a strong influence of $CH_4$ oxidation on the isotopic signature as an enrichment of up to 10‰ cannot be explained solely by upward diffusion of $CH_4$, especially
since the scatter in $\delta^{13}C$-$CH_4$ values was associated with the presence of aerenchymous roots. Oxidation of $CH_4$ is a strongly fractionating processes leaving the remaining $CH_4$ in the pore water enriched by ~10‰ while produced $CO_2$ becomes depleted (Chasar et al., 2000; Popp et al., 1999). Accordingly, $\delta^{13}C$-$CO_2$ values were depleted throughout the rhizosphere and became less negative only at greater depths (Fig. 4e). A more negative $\delta^{13}C$-$CO_2$ signature could also be explained by root respiration adding depleted $CO_2$ to the pore water (Corbett et al., 2013). However, values of $\delta^{13}C$-$CO_2$ of less than -30‰ being
substantially lower than the source organic matter material around -26‰ (Table 3) can only be explained by influence of particularly $^{13}C$ depleted carbon from oxidation of $^{13}C$ depleted $CH_4$.

The scatter of $\delta^{13}C$-$CH_4$ values throughout the rhizosphere and the $\delta^{13}C$-$CH_4$ depth pattern (Fig. 4d) thus indicated a coexistence of microsites because enriched $CH_4$ being only a leftover from $CH_4$ oxidation should result in a $\delta^{13}C$-$CH_4$ depth pattern following that of the $\delta^{13}C$-$CO_2$ profile (Hornibrook et al., 2000). The occurrence of small-scale microsites created by
root $O_2$ release into water-saturated (peat) layers has been previously suggested (Colmer, 2003), also for peatlands on the northern hemisphere (Popp et al., 1999; Knorr et al., 2008b; Corbett et al., 2013). In aerobic, more oxidized microsites attached to roots fresh and labile organic matter and $CH_4$ could be rapidly consumed by aerobic respiration and methanotrophy. These aerobic microsites were likely surrounded by anoxic conditions, enabled by a water table fluctuating near the surface. Within these anaerobic microsites labile organic matter from roots could have served as substrate to produce few amounts of $CH_4$ by
acetoclastic methanogenesis. This was indicated by a $^{12}C/^{13}C$ fractionation factor mostly below 1.055 typical for acetoclastic methanogenesis although based on $^{13}C$ isotope fractionation alone this pathway cannot be clearly separated from methanotrophic conditions (Whiticar et al., 1986; Conrad, 2005). Despite $CH_4$ production by the acetoclastic pathway in anaerobic microsites throughout the rhizosphere supports our third hypothesis, net $CH_4$ production was negligible as produced $CH_4$ was largely consumed in aerobic microsites. Therefore, the apparently small $^{12}C/^{13}C$ fractionation observed within the
rhizosphere could have been arisen by both, acetoclastic methanogenesis and methanotrophy. Below 2 m depth $^{12}C/^{13}C$ fractionation indicated a shift to a predominance of hydrogenotrophic methanogenesis and absence of notable methanotrophic activity (Fig. 4f) as $a_c$-values increased with depth towards 1.065 (Whiticar et al., 1986; Conrad, 2005).

High DIC:$CH_4$ ratios mostly exceeding 100 throughout the prevailingly water-saturated rhizosphere suggested a considerable contribution of further respiration processes to $CO_2$ production. While autotrophic $CO_2$ release by root respiration was



supposed to be substantial, $O_2$ release by roots still exceeded consumption, as demonstrated by $O_2$ saturation reaching up to 10%. This $O_2$ supply may thus not only fuel $CH_4$ oxidation but also heterotrophic respiration using molecular oxygen or other electron acceptors regenerated by this $O_2$ input (Colmer, 2003; Mainiero and Kazda; 2005). Besides aerobic respiration in oxidized microsites attached to roots, other likely sources of $CO_2$ could have been dissimilatory sulfate reduction in anaerobic

microsites of surface peat layers down to 0.4 m with sulfate originating e.g. from sea spray (Kleinebecker et al., 2008; Broder et al., 2015). Elevated $H_2$ levels in the rhizosphere indicated high $H_2$ production by fermentation compared to consumption by e.g. sulfate reduction, presumably due to high root litter input and root exudates (Knorr et al., 2009; Estop-Aragonés et al., 2013).

### 4.3 Pools – low emission scenario

Pools exhibited with $0.23 \pm 0.25$ mmol $CH_4$ $m^{-2}$ $d^{-1}$ second lowest emissions of the microforms under study (Fig. 3). Such low emissions measured from pools were somewhat surprising since they have been previously reported to be similar to *Sphagnum* lawns (Fritz et al., 2011) and were expected to be relevant $CH_4$ sources because of prevailing anoxic conditions (Blodau, 2002). Moreover, other studies identified pools as hot-spots of $CH_4$ emissions from peatlands (Hamilton et al., 1994; Pelletier et al., 2014; Burger et al., 2016). This was clearly not the case in our study, as emissions were in the same order of magnitude as

pool fluxes in South Patagonian *Sphagnum* bogs (Lehmann et al., 2016).

Low emissions from pools were associated with $CH_4$ pore water concentrations following those below *Astelia* lawns on slightly elevated levels (Fig. 4a). These results verified our second hypothesis stating that pore water $CH_4$ concentrations of non-rooted peat below pools remains less affected by $CH_4$ oxidation compared to *Astelia* lawns. Nevertheless, it was surprising that $CH_4$ concentration profiles below pools clearly resembled profiles obtained under *Astelia* lawns. Such concentration profile

suggests that *A. pumila* roots controlled even $CH_4$ dynamics below adjacent microforms as visualized on a conceptual level in Figure 6. To explain low $CH_4$ concentrations in upper peat layers below pools, we suggest that the thorough $CH_4$ oxidation below *Astelia* lawns can also establish a lateral concentration gradient. Thereby, $CH_4$ from adjacent microforms gets redistributed to the rhizosphere of *Astelia* lawns where it is consumed. Throughout the rhizosphere of *Astelia* lawns and in corresponding depths below pools, we observed a highly decomposed and amorphous peat as previously described for cushion

bogs (Ruthsatz and Villagran, 1991; Fritz et al., 2011) indicating a very high water permeability that would together with a high water table position enable such lateral redistribution (Baird et al., 2016). The assumption of lateral pore water movement was supported by a peak in DIC:$CH_4$ ratios accompanied by very low $CH_4$ concentrations around 0.8 m below pools (Fig. 4c) as upward diffusion against the concentration gradient is impossible. Lateral root ingrowth and $O_2$ supply from adjacent *Astelia* lawns could furthermore keep $CH_4$ concentrations in upper peat layers below pools at a low level, in particular if pools are

small.

As observed under *Astelia* lawns, $CH_4$ was produced in deep peat layers below pools also predominantly by hydrogenotrophic methanogenesis and diffused upwards. In upper peat layers influenced by lateral movement of pore water, oxidation of $CH_4$



seemed to occur. However, the sink strength was likely of minor importance compared to *Astelia* lawns, as indicated by less depleted $\delta^{13}$C-$CO_2$ signatures throughout the whole profile (Fig. 4e) on typical levels for peat pore water concentrations (Hornibrook et al., 2000; Beer et al., 2008) and a less enriched $\delta^{13}$C-$CH_4$ signature following largely that of $\delta^{13}$C-$CO_2$ (Fig. 4d, e) indicative for hydrogenotrophic methanogenesis.

Levels of $H_2$ concentrations ~10 nmol $L^{-1}$ sufficient to maintain hydrogenotrophic methanogenesis (Heimann et al., 2010) accompanied by lower DIC:$CH_4$ ratios suggested methanogenic conditions over the whole profile at least in anaerobic microsites. However, overall $CH_4$ production seemed to be limited throughout the profile, especially at peaking $H_2$ levels from 1 to 2 m depth, by low substrate supply from highly decomposed organic matter and thus unfavourable thermodynamic conditions as suggested by very negative $\delta^{13}$C-$CH_4$ signatures (Fig. 4d, 5a) (Knorr et al., 2008a; Hornibrook et al., 1997).

Within the open water of pools which was well-mixed by strong winds $CH_4$ production and excess of $H_2$ was diminished due to electron acceptor availability for consumption (e.g. $O_2$, sulfate). Pool sediments were inhabited by cyanobacteria (Arsenault et al., 2018) and submerged *Sphagnum* mosses (Kip et al., 2012) that provided recalcitrant organic matter for $CH_4$ production in small quantities, as indicated by a slight peak of $CH_4$ concentrations associated with lower DIC:$CH_4$ ratios and elevated $H_2$ ratios around 0.3 m depth. Hydrogenotrophic methanogenesis prevailed as indicated by highly depleted $\delta^{13}$C-$CH_4$ accompanied

with peaking $a_c$-values in that depth (Fig. 4f). Nevertheless, $CH_4$ could not enter to the atmosphere as it was rapidly consumed by methanotrophic bacteria inhabiting *Sphagnum* mosses (Kip et al., 2012; Knoblauch et al., 2015) or associated with cyanobacteria. Thus, the pools at our study site belonged to examples of low emission pools as described in Knoblauch et al. (2015).

**4.4 *Sphagnum* and *Donatia* lawns – low to moderate emission scenario**

$CH_4$ emissions from *Sphagnum* and *Donatia* lawns were most variable among microforms ranging from near-zero to local emission hotspots. *Sphagnum* lawns showed significantly higher $CH_4$ emissions of $1.52 \pm 1.10$ mmol $m^{-2}$ $d^{-1}$ than all other microforms (Fig. 3). Substantially higher fluxes from *Sphagnum* lawns at our study site have been previously described by Fritz et al. (2011) and emissions reported here were in the same order of magnitude as determined for *Sphagnum* lawns in South Patagonian bogs dominated by *Sphagnum* mosses (Broder et al., 2015; Lehmann et al., 2016). Highest emissions

obtained in the present study from *Sphagnum* and *Donatia* lawns were at intermediate levels compared to northern bogs (Blodau, 2002; Laine et al., 2007; Limpens et al., 2008).

We present the first $CH_4$ emission data obtained from *Donatia* lawns. Contrary to our expectations, *Donatia* lawns emitted $0.66 \pm 0.96$ mmol $CH_4$ $m^{-2}$ $d^{-1}$ and had thus second highest $CH_4$ emissions of our study. One lawn even exceeded emissions of the strongest *Sphagnum* lawn source (Fig. 3c, Table 2) on all measurement occasions. As these comparatively high emissions

could be well reproduced, we conclude that an artefact can be excluded. Therefore, our first hypotheses, that cushion plants emit negligible amounts of $CH_4$, could retrospectively only be partly confirmed, as it depends on the specific cushion forming plant species and likely its root characteristics. Cushion-forming *D. fascicularis* seemed to be more comparable to *Sphagnum*



lawns regarding $CH_4$ emissions. We conclude that the presence of cushion plants as a proxy for negligible $CH_4$ emissions from cushion bogs needs to be carefully interpreted.

*D. fascicularis* is like *A. pumila* a cushion-forming vascular plant establishing a deep and aerenchymous, but with < 0.05 g DW L$^{-1}$ a substantially less dense rooting system compared to *A. pumila* with > 4 g DW L$^{-1}$ in around 0.5 m depth (Fritz et al.,

2011). The lower root density of *D. fascicularis* was probably below a specific threshold at which root $O_2$ release did not suppress $CH_4$ emission any more. Instead, the roots might have promoted $CH_4$ production by presence of labile root organic matter and even functioned as conduits for gas transport from anoxic peat layers to the atmosphere thereby enhancing $CH_4$ emissions (e.g. Joabsson et al., 1999; Colmer, 2003; Whalen, 2005). As root $O_2$ release facilitates nutrient mobilization (Colmer, 2003), this assumption of little if any $O_2$ release by *D. fascicularis* roots is in line with Schmidt et al. (2010) who

found nutrient concentrations in *D. fascicularis* to be almost as low as in *S. magellanicum* while *A. pumila* contained significantly higher amounts of nutrients.

$CH_4$ production, consumption and emission in the study site appeared to be a function of root characteristics. Above a specific threshold of root density and activity associated with $O_2$ release, $CH_4$ emissions were suppressed while below this threshold, $CH_4$ production was accelerated by presence of labile root organic matter or exudates accompanied with $O_2$ consumption along

with enhanced $CH_4$ transport by aerenchymatic roots (Blodau, 2002; Agethen et al., 2018). Such relationship was suggested by our preliminary observations of microform features associated with the pronounced spatial variability of $CH_4$ emissions and would explain why the *A. pumila* root oxygenation impact showed a pronounced spatial pattern and did not control the $CH_4$ dynamics in the entire upper peat. Accordingly, *Sphagnum* and *Donatia* lawns that showed substantial $CH_4$ emissions were beyond the sphere of influence by *A. pumila* roots. Although not measured here, $CH_4$ concentrations in the pore water

below those lawns probably increased steeply below the water table (Fig. 6), like common depth profiles in northern (e.g. Beer and Blodau, 2007; Limpens et al., 2008; Corbett et al., 2013) or southern bogs (Broder et al., 2015). This explanation implies that the pore water below those lawns with remarkably higher emissions was not well-connected to pore water influenced by *A. pumila* roots. A spatial variability of peat physical properties on the ecosystem scale was suggested by water table records (Fig. 2) demonstrating that one site responded faster than the other to, for instance, precipitation events.

We speculate that the biogeochemistry below *Sphagnum* and *Donatia* lawns with near-zero emissions was, likewise to pools, influenced by the pronounced root activity of nearby *A. pumila* establishing a lateral concentration gradient for $CH_4$ thereby reducing $CH_4$ emissions (as visualized in Fig. 6). This assumption was supported by previous research in our study site. Fritz et al. (2011) showed that below *Sphagnum* lawns pore water $CH_4$ concentrations can be substantially lowered throughout rhizosphere-influenced peat layers compared to a reference site without *A. pumila* while in deep peat layers $CH_4$ concentrations

increased steeply. Further research should be undertaken to investigate why $CH_4$ emissions from *Sphagnum* and *Donatia* lawns appear to be highly variable in order to understand $CH_4$ dynamics in under-researched cushion bogs.





### 4.5 Implications for ecosystem CH₄ emissions from Patagonian cushion bogs

Highest summer emissions in the present study were considerably lower compared to maximum austral summer fluxes of ~150 mg CH$_4$ m$^{-2}$ d$^{-1}$ (9.4 mmol m$^{-2}$ d$^{-1}$) determined for a New Zealand bog dominated by the vascular evergreen "wire rush," *Empodisma robustum* (Goodrich et al., 2015). The authors explained such high CH$_4$ emissions by comparatively wet conditions

and a high density of aerenchymous vegetation providing a gas conduit for CH$_4$ transport in their study site. Compared to CH$_4$ emissions of northern bogs ranging from 3 to 53 mg m$^{-2}$ d$^{-1}$ at intermediate levels in wet bogs to up to 80 mg m$^{-2}$ d$^{-1}$ (Blodau, 2002; Laine et al., 2007; Limpens et al., 2008; 0.2 to 2.2 and up to 5 mmol m$^{-2}$ d$^{-1}$), highest summer emissions in the present study represent a pronounced CH$_4$ source. Nevertheless, overall ecosystem CH$_4$ emissions would probably be among the lowest described for pristine bog ecosystems worldwide when taking into account the high proportion of the surface area where *Astelia*

lawns prevailed.

While *Donatia* lawns covered only small parts of our study site, we observed other parts of the complex bog system in the whole study area to be widely dominated by *D. fascicularis* instead of *A. pumila* which is in line with previous studies (Heusser, 1995; Grootjans et al., 2014) who described the cushion bogs in the Moat landscape to be covered by both, *A. pumila* and *D. fascicularis*. Thus, amounts of emitted CH$_4$ could be significant also on the ecosystem scale once *D. fascicularis* dominates.

For instance, a predominance of *D. fascicularis* is typical for later vegetation succession stages in cushion bogs of Tierra del Fuego (Heusser, 1995) or Chile (Ruthsatz and Villagran, 1991; Kleinebecker et al., 2007).

### 5 Conclusion

We conclude from our study that root density and activity of cushion-forming *A. pumila* and associated O$_2$ supply strongly controlled CH$_4$ production and consumption in a pristine Patagonian cushion bog. Thereby, CH$_4$ emission were reduced to

near-zero levels and largely decoupled from environmental controls. The high root density even regulated CH$_4$ dynamics below adjacent microforms with less or non-rooted peat such as pools, *Sphagnum* lawns or lawns dominated by cushion-forming *D. fascicularis* by maintaining lateral concentration gradients in upper peat layers. Nevertheless, when root density dropped below a certain threshold, CH$_4$ production might have been accelerated by presence of labile root organic matter or exudates accompanied with O$_2$ consumption along with enhanced CH$_4$ transport by aerenchymatic roots of e.g. *D. fascicularis*. Under

such circumstances, CH$_4$ emissions increased to intermediate levels compared to northern bogs. Therefore, the presence of cushion plants as a proxy for negligible CH$_4$ emissions from cushion bogs should be interpreted carefully. As cushion bogs can be found in many (mostly southern) parts of the world and only very limited knowledge about CH$_4$ dynamics from these systems exist, future research should take into account a possibly high spatial variability of CH$_4$ emission from bogs dominated by cushion plants. We demonstrated an extreme scenario for how a spatial distribution of root density in the peat can lead to a

pronounced pattern of CH$_4$ emissions. Yet, the underlying ratio between root characteristics and O$_2$ supply determining this emission pattern should be applicable to other densely rooted peatlands in general. Further research should be undertaken to





prove relationships that have been developed here on a conceptual level in order to extend our knowledge on $CH_4$ dynamics in under-researched cushion bogs.

*Data availability.* The data can be accessed by email request to the corresponding authors.

5 *Author contributions.* CB, TK and WM designed the study. WM, CB, TK and VAP conducted field work and sample analyses with the help of KHK. WM performed data analyses and prepared the manuscript with contributions from KHK, TK and VAP.

*Competing interests.* The authors declare that they have no conflict of interest.

*Acknowledgement.* This study was carried out within the research project CANDYbog which was funded by the Deutsche
10 Forschungsgemeinschaft (German Research Foundation, DFG; Grant No. KL2265/3-1 and BL 563/19-1). We highly appreciated the Centro Austral de Investigaciones Científicas (CADIC-CONICET) providing facilities that enabled our fieldwork in Ushuaia. The Prefectura Naval Argentina is acknowledged for offering facilities supporting our fieldwork. Lucas Varela of the La Posta Hostel family is appreciated for offering working space and logistics. We wish to acknowledge the laborious field assistance of Isabella Närdemann, Carla Bockermann, Claudia Frank, Bettina Breuer, Lina Birkner, Juliane
15 Kohlstruck, Laura Jansen, David Holl, Lars Kutzbach, María Florencia Castagnani and María Noel Szudruk Pascual. All analyses of this study were carried out in the laboratory of the institute of Landscape Ecology. The assistance of Ulrike Berning-Mader, Sebastian R. Schmidt, Ronya Wallis, Sabrina Knaack, Sina Berger, Simona Bonaiuti and Leandra Praetzel is greatly acknowledged. We thank Peter Sulmann for technical assistance. The support by Open Access Publication Fund of University of Muenster is acknowledged.



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





**Figures**

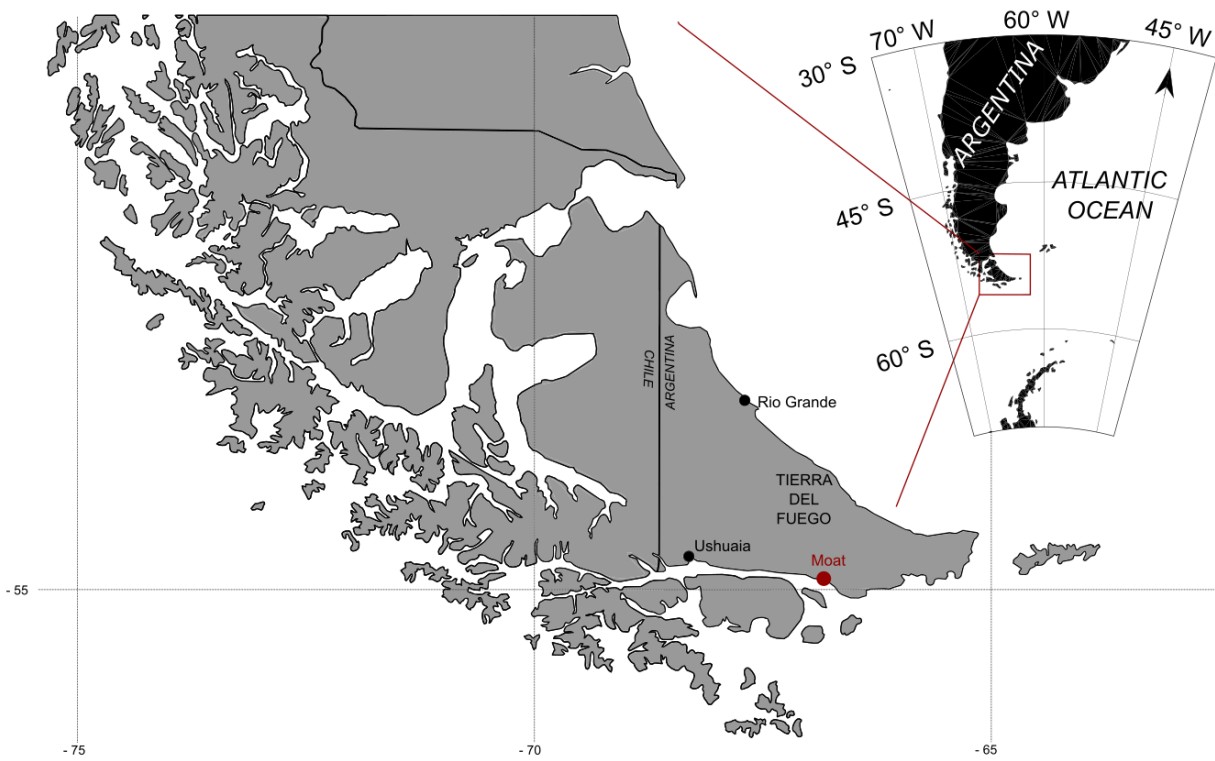

**Figure 1.** Location of the study area in southernmost Patagonia. The investigated cushion bog in Moat is located 130 km eastern of Ushuaia (Argentina). The map was created at http://www.simplemappr.net accessed on 11.12.2017 and in Matlab
5 (MATLAB Mapping Toolbox Release R2017a).



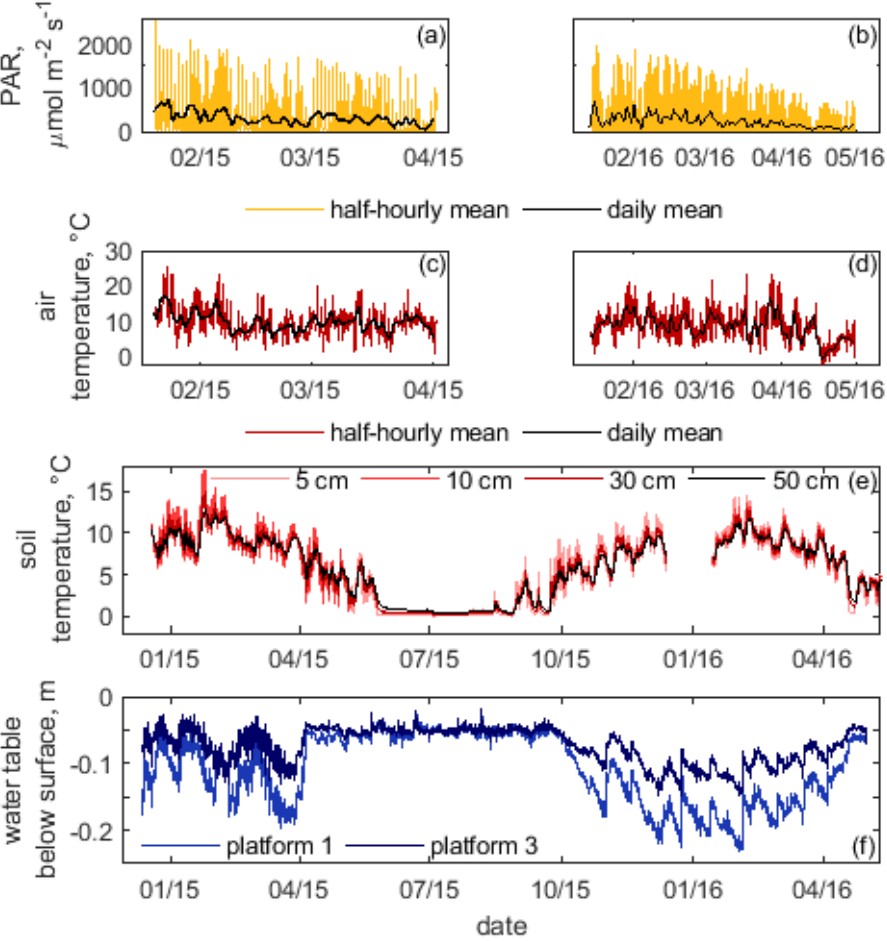

**Figure 2.** Time series of environmental variables during the study period. Photosynthetic active radiation (PAR, a, b) and air temperature (c, d) are presented as half-hourly and daily means during two austral summer periods. Soil temperature (e) and water table position below surface (f) were continuously recorded and are shown as half-hourly means. Note the difference in response time between platform 1 and platform 2 where water table fluctuations were recorded.





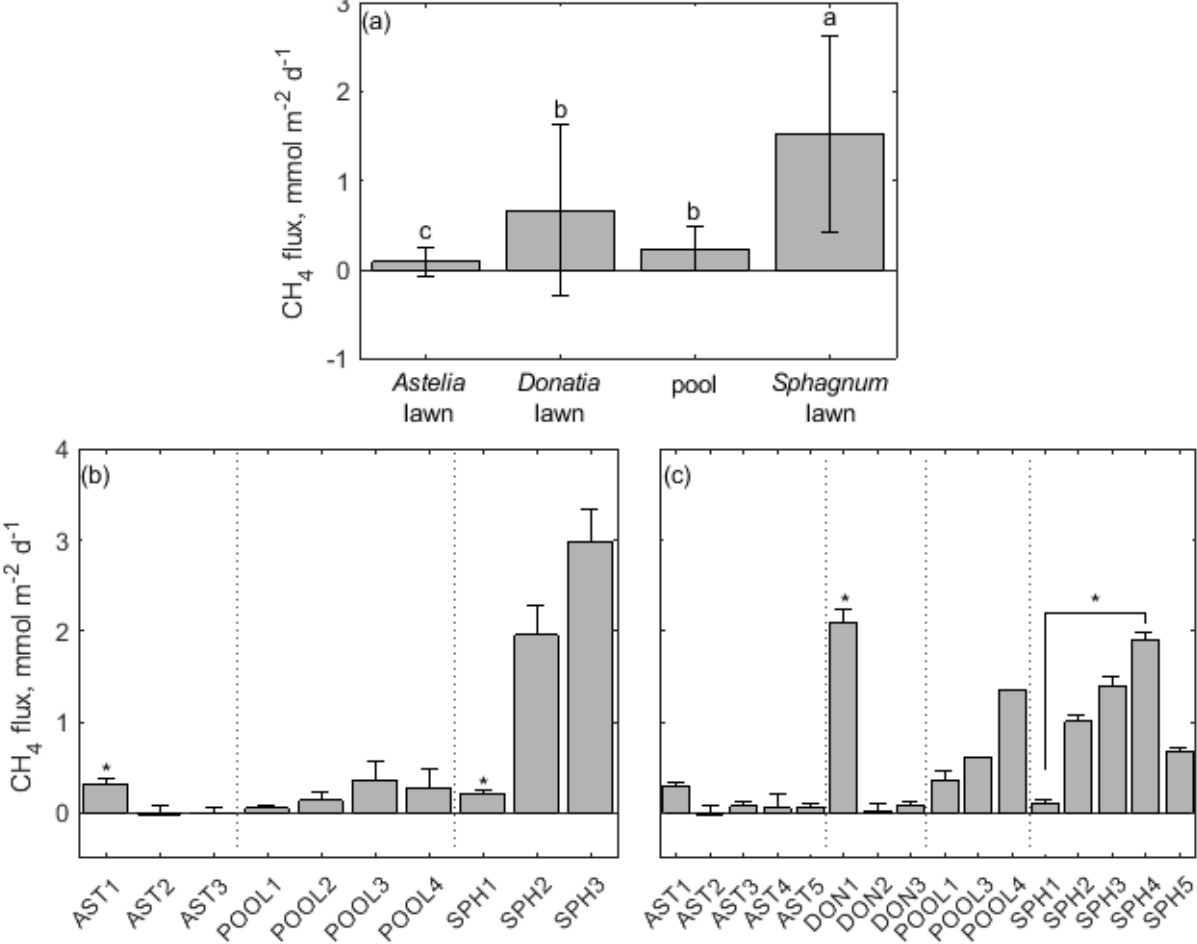

**Figure 3 a-c.** Mean $CH_4$ emissions and their standard deviation determined from dominant microforms (a) and individual collars during the two measurement campaigns (b, c) in a Patagonian cushion bog. Different letter superscripts denote significantly different amounts of emitted $CH_4$ between microforms despite variability within microforms was substantial (a) or from collars within one microform (b, c) that were identified by Kruskal-Wallis analysis of variance and multiple comparison U-tests with Bonferroni adjustment.







**Figure 4 a-f.** Pore water composition in depth profiles obtained from MLPS (N = 3) installed in *Astelia* lawns and pools during two sampling campaigns in austral summer 2015 and 2016. Mean values and their standard deviation are presented. CH₄ (a) and DIC (b) concentrations as well as related DIC:CH₄ (c) ratios were determined during both sampling events while carbon

5  isotope values of CH₄ (d) and DIC (e) and corresponding fractionation factors (f) were investigated only during the second





campaign. DIC:CH$_4$ ratios are shown on a logarithmic scale without negative standard deviation for *Astelia* lawns and pools. Displayed depths represent the centre of a MLP segment and had to be standardized because of different sampling resolution in both years. The water table position and approximate maximum rooting depth (rhizosphere) refer to *Astelia* lawns only. Note the different scale of the y-axis in the upper and lower three panels.

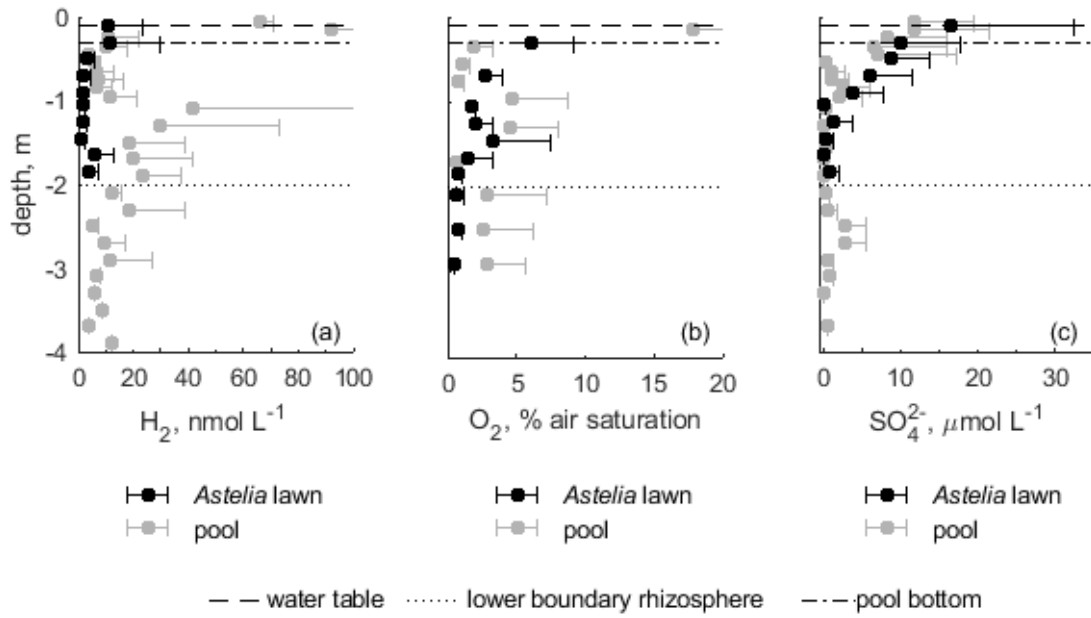

**Figure 5 a-c.** Profiles of H$_2$ (a), O$_2$ (b) and sulfate (c) pore water concentrations obtained from MLPs (N = 3) installed in *Astelia* lawns and pools during two sampling campaigns in austral summer 2015 and 2016. Mean values and their standard deviation are presented. Displayed depths represent the centre of a MLP segment and had to be standardized because of different sampling resolution in both years. The water table position and approximate maximum rooting depth (rhizosphere)

10   refer to *Astelia* lawns only.





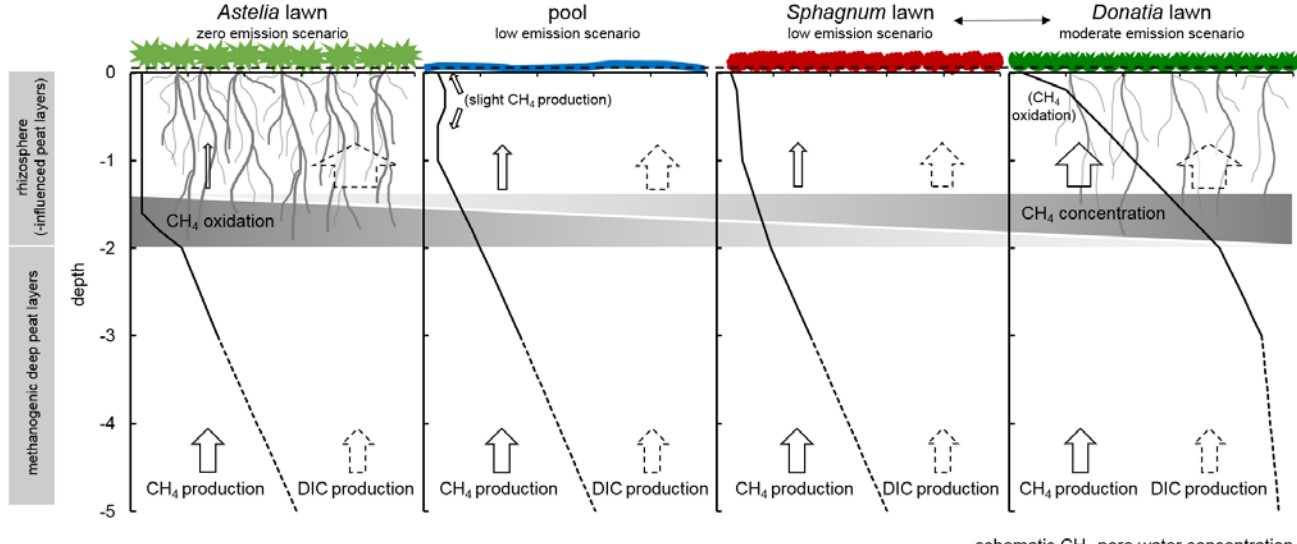

**Figure 6.** Schematic CH$_4$ concentration profiles below the four dominant microforms in a Patagonian cushion bog. The shape of profiles was derived from data obtained in the present study together with those by Limpens et al. (2008), Fritz et al. (2011) and Broder et al. (2015). The size of arrows represents the relative magnitude of either a CH$_4$ or CO$_2$ flux following the diffusion gradient. The water table position is displayed by a dashed line near the surface. CH$_4$ oxidation by root O$_2$ release resulted in near-zero CH$_4$ pore water concentrations in densely rooted peat below *Astelia* lawns. With increasing distance from densely rooted peat, CH$_4$ oxidation decreased while CH$_4$ concentration increased until reaching a depth profile typical for those of northern peatlands with less if any roots in water-saturated peat layers. Exemplary sequence of microforms, derived from observed amounts of emitted CH$_4$. Other sequences of microforms are reasonable as microforms occur in various combinations in the field.





**Tables**

Table 1. Characteristic plant species of dominant microforms in a Patagonian cushion bog. Species composition was determined during the second measurement campaign within collars where closed chamber measurements were conducted. Mean cover (%) and its standard deviation of characteristic were given and plant species with a mean cover < 5 % were listed with +.

|  | microform | | |
| --- | --- | --- | --- |
| species | *Astelia* lawn | *Sphagnum* lawn | *Donatia* lawn |
| *Astelia pumila* | 84 ± 16 | | |
| *Donatia fascicularis* | 8 ± 10 | + | 90 ± 9 |
| *Sphagnum magellanicum* | | 94 ± 4 | |
| *Tetroncium magellanicum* | + | 7 ± 2 | + |
| *Caltha dioneifolia* | + | + | 6 ± 5 |
| *Empetrum rubrum* | + | | |
| *Gaultheria antarctica* | + | + | + |
| *Myrteola nummularia* | + | | |



Table 2. $CH_4$ fluxes determined from individual collars installed in dominant microforms of a Patagonian cushion bog. Number of flux measurements, their mean and standard deviation are presented. Significantly different fluxes from collars within one microform were identified by Kruskal-Wallis analysis of variance and multiple comparison U-tests with Bonferroni adjustment. Results of computing Spearman's rank correlation coefficient rho for the relationships between $CH_4$ fluxes and possible environmental controls, namely water table depth determined separately for each collar and soil temperature in 0.05 m depth, are given. Furthermore, characteristics of collars and results of computing Spearman's rank correlation for the relationships between mean $CH_4$ fluxes during the second campaign and each feature are shown. Microform size refers to the extent of a microform in which a specific collar was installed. P-values < 0.05 indicate a correlation significantly different from zero.





| year | micro-form | | N | Kruskal-Wallis analysis — mean flux, mmol CH$_4$ m$^{-2}$ d$^{-1}$ | p | water table depth, m — rho | p | soil temperature, °C — rho | p | presence *T. magellanicum* — number of shoots | rho | p | presence *D. fascicularis* — cover, % | rho | p | microform size — size, m² | rho | p |
|---|---|---|---|---|---|---|---|---|---|---|---|---|---|---|---|---|---|---|
| 2015 | *Astelia* lawn | 1 | 60 | 0.32 ± 0.05 | < 0.05 | -0.22 | n.s. | n.d. | n.d. | | | | | | | | | |
| | | 2 | 64 | -0.02 ± 0.09 | n.s. | -0.21 | n.s. | 0.17 | n.s. | | | | | | | | | |
| | | 3 | 55 | -0.01 ± 0.07 | n.s. | 0.03 | n.s. | -0.08 | n.s. | | | | | | | | | |
| | *Sphagnum* lawn | 1 | 62 | 0.21 ± 0.05 | < 0.05 | -0.21 | n.s. | n.d. | n.d. | | | | | | | | | |
| | | 2 | 64 | 1.96 ± 0.33 | n.s. | 0.41 | < 0.05 | 0.52 | < 0.05 | | | | | | | | | |
| | | 3 | 54 | 2.99 ± 0.35 | n.s. | 0.2 | n.s. | 0.6 | < 0.05 | | | | | | | | | |
| | pool | 1 | 13 | 0.05 ± 0.03 | n.s. | | | n.d. | n.d. | | | | | | | | | |
| | | 2 | 12 | 0.14 ± 0.08 | n.s. | | | 0.4 | n.s. | | | | | | | | | |
| | | 3 | 10 | 0.36 ± 0.2 | n.s. | | | 0.64 | < 0.05 | | | | | | | | | |
| | | 4 | 11 | 0.28 ± 0.2 | n.s. | | | -0.01 | n.s. | | | | | | | | | |
| 2016 | *Astelia* lawn | 1 | 7 | 0.29 ± 0.04 | n.s. | 0.27 | n.s. | -0.13 | n.s. | 15 | | | 25 | | | 1.12 | | |
| | | 2 | 11 | -0.02 ± 0.1 | n.s. | -0.22 | n.s. | -0.16 | n.s. | 0 | | | 4 | | | 4.45 | | |
| | | 3 | 8 | 0.07 ± 0.05 | n.s. | -0.68 | n.s. | -0.31 | n.s. | 0 | 0.67[a] | n.s. | 0 | 0.01 | n.s. | 26.66 | -0.10 | n.s. |
| | | 4 | 11 | 0.06 ± 0.15 | n.s. | -0.09 | n.s. | 0.13 | n.s. | 0 | | | 8 | | | 3.38 | | |
| | | 5 | 11 | 0.06 ± 0.04 | n.s. | -0.44 | n.s. | 0.14 | n.s. | 6 | | | 2 | | | 4.55 | | |
| | *Donatia* lawn | 1 | 9 | 2.1 ± 0.14 | < 0.05 | -0.82 | < 0.05 | 0.74 | < 0.05 | 30 | | | 80 | | | 0.26 | | |
| | | 2 | 11 | 0.01 ± 0.09 | n.s. | 0.29 | n.s. | -0.64 | < 0.05 | 9 | 1.00[a] | n.s. | 97 | -1.00[a] | n.s. | 0.15 | 0.50 | n.s. |
| | | 3 | 10 | 0.08 ± 0.04 | n.s. | -0.46 | n.s. | -0.2 | n.s. | 18 | | | 92 | | | 0.96 | | |
| | *Sphagnum* lawn | 1 | 9 | 0.09 ± 0.05 | < 0.05 | 0.68 | n.s. | -0.78 | < 0.05 | 66 | 0.40 | n.s. | 0 | 0.71[a] | n.s. | 0.43 | 0.00 | n.s. |
| | | 2 | 11 | 1.01 ± 0.06 | n.s. | 0.15 | n.s. | -0.42 | n.s. | 165 | | | 0 | | | 2.38 | | |





|  | 3 | 11 | 1.39 ± 0.1 | n.s. | -0.05 | n.s. | -0.05 | n.s. | 175 | 0 | 2.7 |
|  | 4 | 9 | 1.9 ± 0.08 | < 0.05 | -0.44 | n.s. | 0.25 | n.s. | 108 | 4 | 0.38 |
|  | 5 | 10 | 0.68 ± 0.03 | n.s. | 0.17 | n.s. | -0.22 | n.s. | 120 | 0 | 0.65 |
| pool | 1 | 2 | 0.35 ± 0.11 | n.s. |  |  | n.d. | n.d. |  |  |  |
| pool | 3 | 1 | 0.6 ± 0 | n.s. |  |  | n.d. | n.d. |  |  |  |
| pool | 4 | 1 | 1.35 ± 0 | n.s. |  |  | n.d. | n.d. |  |  |  |

n.s. = not significant, n.d. = not determined, [a] = despite a (relative) high absolute value of Spearman's rho, p-values did not indicate a significant correlation, probably because of outliers.





Table 3. Summary of peat characteristics obtained from two peat cores taken in *Astelia* lawns in a Patagonian cushion bog. Data were averaged over the respective depths sampled in 0.1 m resolution.

| peat characteristics | depth, m | bulk density, g cm$^{-3}$ | N, % | C, % | CN | $\delta^{15}$N | $\delta^{13}$C |
|---|---|---|---|---|---|---|---|
| amorphous, highly decomposed cushion peat with many living *Astelia* roots, only close to the surface (0-0.4 m) with recognizable plant material | 0-1 | 0.03 | 1.42 | 51.07 | -1.13 | -25.65 | 38.76 |
| | 1-2 | 0.06 | 1.83 | 55.51 | 1.24 | -26.18 | 30.37 |
| *Sphagnum* peat, with depth increasingly amorphous and decomposed | 2-3 | 0.06 | 1.52 | 53.95 | 1.12 | -26.33 | 36.19 |
| | 3-4 | 0.06 | 1.22 | 50.75 | 0.22 | -24.55 | 42.24 |
| | 4-5 | 0.11 | 1.17 | 47.86 | -1.59 | -26.43 | 41.45 |
| | 5-6 | 0.10 | 1.18 | 47.56 | -1.62 | -26.77 | 40.67 |
| | 6-7 | 0.15 | 1.28 | 50.27 | -1.97 | -27.08 | 44.34 |