# Peer review of "Zero to moderate methane emissions in a densely rooted, pristine Patagonian bog - biogeochemical controls as revealed from isotopic evidence"

_Biogeosciences, 2018_

## Referee Comment (RC1) · Anonymous Referee #1 · 18 Sep 2018

This is a carefully done study about the production, oxidation and emission of CH4 in Patagonian bog, the results are of considerable interest and the paper is well written. However, some points need clarifying and certain statements require further justification. 1. the authors should not ignore that acetogenesis might be important in anaerobic environments when H2 partial pressures are high and temperatures are low. Acetogens can outcompete methanogens at low temperature, as many acetogens seem to have a higher growth rate at low temperature than most methanogens (Kotsyurbenko et al., 1996, 2001). If acetogenesis process is active in the bog, the $\delta$13C value of

actetate in the porewater will be largely decreased because of the substantial fractionation during acetate production from CO2 and H2. And resultantly, the 13C value of CH4 will also be lower and resulted in larger apparent isotopic fractionation factor (ac) between CO2 and CH4. Therefore, it's difficult to determine the relative importance of acetoclastic versus hydrogenotrophic methanogenesis pathway without the 13C value of acetate in this study. 2. In the first page, line 26-28, it's stated that: "Below the rhizosphere. . . . . .CH4 was predominantly produced by hydrogenotrophic methanogenesis". In fact, data in Figure 4def showed that the hydrogenotrophic pathway had higher contribution to CH4 in the pool, while the acetoclastic pathway must play relatively more important role for the CH4 production below the rhizosphere of Astelia Lawn. This is consistent with the supply of labile organic carbon from the root exudates of Astelia. To sum up, I think it's difficult to conclude that CH4 is mainly produced from the hydrogenotrophic pathway below the rhizosphere of Astelia. 3. It's stated that mean root lifetimes of A. pumila has been estimated to be ∼3-4 years. So, whether the production and oxidation of CH4 will be strongly affected in case of the turnover of large amounts of roots? 4. Please check Table 3, the data in the last three columns are in wrong places.

---

## Referee Comment (RC2) · Anonymous Referee #2 · 27 Sep 2018

Comments to the ms bg-2018-301

"Zero to moderate methane emissions in a densely rooted, pristine Patagonian bog - biogeochemical controls as revealed from isotopic evidence"

General comments The MS of Münchberger and co-authors contributes to the knowledge on processes of methane turnover (production+oxidation) and transport in rarely studied southern bogs. Authors combined field sampling with the advanced analytics (porewater chemistry and stable isotope analyses) to report relationships and peculiar mechanisms between microrelief forms, dominating vegetation communities and the net processes affecting the CH4 efflux from the Patagonian peatland during two consecutive summer seasons. Field-based studies are critically important for understanding processes related to functioning of ecosystems and therefore interesting for the broad scientific community. Accepting the field experiments typically operate with much larger spatial and temporal variability in measured parameters (and as the result, relatively lower statistical power as compared to controlled conditions), still there are several issues which I would like to point out for the discussion and improvement. Below authors find general comments while specific recommendations and technical corrections are incorporated directly in the draft file attached. 1. First of all, the MS is rather long and too repetitive and descriptive. Thus, the Introduction is definitely too extended, especially regarding the common knowledge about methane in the very beginning and peatlands in general. Authors could immediately start the story of the importance of southern peatlands and have the necessary information on peatlands' biochemistry and vegetation specialty in there. Then the information on the isotope issue would be sufficient to formulate hypotheses without any loss of logic. 2. In the proposed hypotheses, it has to be clearer why pools are so much different from lawns in terms of methanogenesis pathways. This was not strait forward from the introduction; I suggest to omit statements as "remains less affected" because they are more confusing then explanatory; please, rephrase. 3. In the Methods section, I was confused with relatively short time (3 min) of chamber exposition even under the conditions of rather low atmospheric temperatures and low fluxes expected. Why also transparent and not opaque chambers were used for CH4 fluxes measurement? 4. Discussion section contains repetitive and partly speculative information and therefore is currently too long. For instance, in the discussion of results on 13C-CH4 depth profile (page 14, lines 5-8) authors seemingly "oversell" their results: "scattered between" may also indicate no significant difference (this is not clear from the data). Indeed, Fig. 4d demonstrates rather narrow d13C-CH4 range along the whole depth profile. So, in fact, d13C-CH4 signal alone was not informative enough to approve the strong oxidative properties of rhizosphere of A. pumila. I agree that both methanogenesis and oxidation may co-exist in close vicinity, but still it may not explain lack of d13C-CH4 variation between upper and lower horizons unless CH4 produced in the rhizosphere region is even more depleted in 13C than in deeper layers. The explanation of this phenomenon because of "more reduced...microsites" is not fully clear. More than below the rhizosphere? Why? 5. Contribution of acetoclastic pathway to methanogenesis in the rhizosphere of A. pumila was not convincingly verified (e.g. page 14, lines 23-27) and looks therefore speculative: having acetoclastic methanogenesis and co-existence of oxidation should generate much more enriched d13C-CH4 values in comparison to deep peat. Fig. 4d cannot support this. Seemingly, change of fractionation factor with depth was not significant either. The available data are not enough to approve existence of acetoclastic methanogenesis, and this has to be acknowledged. 6. Another critical point is again a speculative discussion of the results on pools and lateral flows on the site (page 15, lines 23-35). Explanations on gas diffusion along gradient were clear for me (from pools to lawns) but water movement is not the same. Pools are local depressions, so water should flow from lawns into pools. If this flow is so low, then the gas diffusion in opposite direction can be stronger, but this means almost standing water. In case there is a lateral flow of water (what is very natural), then the gas flow can't be counter to it. Therefore, I could understand the inflow of oxygen from lawns into pools, but not CH4 from pools to lawns. The overall picture may change if there is a slope, but then lawns and pools have to be arranged accordingly. Pools will get matter of those lawns which are exposed higher and transfer it downwards to other lawns. If there is a slope on the site, then the conceptual figure should somehow reflect it. Such important information was not provided in Mat&Meth or any other parts of MS. 7. The section 4.4. is rather long and at several places contains repetitive text (e.g. page 17, lines 15-17, 21-23, 27-28; the effect of A. pumila roots was very clear, no need to repeat many times). I recommend condensing text strongly. 8. Depending on the available information from authors, the conceptual Fig. 6 can be changed (see more detailed comments in the text).

Please also note the supplement to this comment:
https://www.biogeosciences-discuss.net/bg-2018-301/bg-2018-301-RC2-
supplement.pdf

**Supplement:**

**Zero to moderate methane emissions in a densely rooted, pristine Patagonian bog - biogeochemical controls as revealed from isotopic evidence**

Wiebke Münchberger[1, 2], Klaus-Holger Knorr[1], Christian Blodau[1, †], Verónica A. Pancotto[3, 4], Till Kleinebecker[2]

[1]Ecohydrology and Biogeochemistry Research Group, Institute of Landscape Ecology, University of Muenster, Heisenbergstraße 2, 48149 Muenster, Germany
[2]Biodiversity and Ecosystem Research Group, Institute of Landscape Ecology, University of Muenster, Heisenbergstraße 2, 48149 Muenster, Germany
[3]Centro Austral de Investigaciones Científicas (CADIC-CONICET), B. Houssay 200, 9410 Ushuaia, Tierra del Fuego, Argentina
[4]Instituto de Ciencias Polares y Ambiente (ICPA-UNTDF), Fuegia Basket, 9410 Ushuaia, Tierra del Fuego, Argentina

*Correspondence to*: Wiebke Münchberger (wiebke.muenchberger@uni-muenster.de), Klaus-Holger Knorr (kh.knorr@uni-muenster.de)

**Abstract.** Peatlands are significant global methane ($CH_4$) sources, but processes governing $CH_4$ dynamics have been predominantly studied on the northern hemisphere. Southern hemispheric and tropical bogs can be dominated by cushion-forming vascular plants (e.g. *Astelia pumila*, *Donatia fascicularis*). These cushion bogs are found in many (mostly southern) parts of the world but could also serve as extreme examples for densely rooted northern hemispheric bogs dominated by rushes and sedges. We report highly variable summer $CH_4$ emissions from different microforms in a Patagonian cushion bog as determined by chamber measurements. Driving biogeochemical processes were identified from pore water profiles and carbon isotopic signatures. An intensive root activity within a rhizosphere stretching over 2 m depth accompanied by molecular oxygen release created aerobic microsites in water-saturated peat leading to a thorough $CH_4$ oxidation ($< 0.003$ mmol $L^{-1}$ pore water $CH_4$, enriched $\delta^{13}C$-$CH_4$ by up to 10‰) and negligible emissions ($0.09 \pm 0.16$ mmol $CH_4$ $m^{-2}$ $d^{-1}$) from *Astelia* lawns. Root activity even suppressed $CH_4$ emissions from non-rooted peat below adjacent pools ($0.23 \pm 0.25$ mmol $CH_4$ $m^{-2}$ $d^{-1}$), in which we found similar pore water profile patterns as obtained under *Astelia* lawns. Below the rhizosphere pore water concentrations increased sharply to $0.40 \pm 0.25$ mmol $CH_4$ $L^{-1}$ and $CH_4$ was predominantly produced by hydrogenotrophic methanogenesis. Few *Sphagnum* lawns and – surprisingly – one lawn dominated by cushion-forming *D. fascicularis* were found to be local $CH_4$ emission hot spots with up to $1.52 \pm 1.10$ mmol $CH_4$ $m^{-2}$ $d^{-1}$ presumably as root density and molecular oxygen release dropped below a certain threshold. The spatial distribution of root characteristics supposedly causing such pronounced $CH_4$ emission pattern was evaluated on a conceptual level that aimed to reflect extreme examples of general scenarios in densely rooted bogs. We conclude that presence of cushion vegetation as a proxy for negligible $CH_4$ emissions from cushion bogs needs to be interpreted with caution. Nevertheless, overall ecosystem $CH_4$ emissions at our study site were

**Summary of comments: bg-2018-301-manuscript-version1.pdf**

**Page:1**

Number: 1  Author:   Subject: Highlight  Date: 2018-09-21 11:36:39

How was this tested? If so, how far from root surface the "suppressive effect" is possible?

Number: 2  Author:   Subject: Highlight  Date: 2018-09-21 11:38:02

Please, rephrase the sentence in a more simplistic way. Too difficult to read and understand.

probably minute compared to bog ecosystems worldwide and widely decoupled from environmental controls due intensive root activity of e.g. *A. pumila*.

**1 Introduction**

Peatland ecosystems are long-term carbon (C) stores as organic C in plant residues degrades only incompletely and
accumulates over millennia (Gorham, 1991; Yu et al., 2012). While being C sinks, peatlands are significant natural methane ($CH_4$) sources on the global scale responsible for about 10% of global annual CH4 emissions (Aselmann and Crutzen, 1989;[1] Mikaloff Fletcher et al., 2004). $CH_4$ has a  higher [2] global warming potential over a 100-year time horizon compared to carbon dioxide ($CO_2$) (IPCC, 2014) and, thus, processes governing $CH_4$ production, consumption and emissions from peatlands receive much attention to estimate global greenhouse gas emissions.

The slowdown of decomposition and accumulation of C in rainwater-fed peatlands (bogs) results from e.g. a high water table, [3] anoxic conditions, recalcitrant peat forming litter, inactivation of oxidative enzymes and an accumulation of decomposition end-products (Beer and Blodau, 2007; Limpens et al., 2008; Bonaiuti et al., 2017). The upper, unsaturated and oxic peat layers with high decomposition rates typically extent only about few decimetres (Whalen, 2005; Limpens et al., 2008). Once methanogenic conditions are established, $CH_4$ production is mainly controlled by substrate supply of degradable organic matter
(Hornibrook et al., 1997; Whalen, 2005). Produced $CH_4$ is released from peat by three main pathways: diffusion, ebullition and plant-mediated transport (Blodau, 2002; Limpens et al., 2008). During slow diffusive transport along the concentration gradient from deep anoxic peat layers to the atmosphere, $CH_4$ might get trapped [4] within upper, oxic zones and consumed by methanotrophic microbes (Chasar et al., 2000; Whalen, 2005; Berger et al., 2018). Consequently, water table position and fluctuations that determine the extent of oxic zones strongly control the amount of emitted $CH_4$ (Blodau and Moore, 2003;
Whalen, 2005). While $CH_4$ oxidation by methanotrophs results in suppressed $CH_4$ emissions, ebullition by fast release of gas bubbles can substantially increase $CH_4$ emissions (Fechner-Levy and Hemond, 1996; Whalen, 2005; Knoblauch et al., 2015; Burger et al., 2016). Many wetland plants establish aerenchymatic root tissues to supply roots growing in anoxic peat with molecular oxygen ($O_2$) (Joabsson et al., 1999; Colmer, 2003). Yet, these aerenchymatic roots have frequently been described to act as conduits to the atmosphere for $CH_4$ transport bypassing oxic surface layers thereby preventing $CH_4$ oxidation and
enhancing emissions (e.g. Joabsson et al., 1999 and references therein; Chasar et al., 2000; Colmer, 2003, Whalen, 2005, Knoblauch et al., 2015; Berger et al., 2018). Vegetated or open water pools that are characteristic of peatlands have received less attention than the vegetated surfaces (Pelletier et al., 2014). However, pools are considered as strong $CH_4$ emitters due to their anoxic conditions (Hamilton et al., 1994; Blodau 2002; Burger et al., 2016) that can even turn the peatlands' C balance into a source (Pelletier et al., 2014) but examples of low emission pools have been also reported (Knoblauch et al., 2015).
Carbon isotopic signatures are a valuable tool for identifying mechanisms and pathways of $CH_4$ production and consumption. [5] Carbon isotopic signatures in pore water of northern hemispheric bogs (hereafter termed northern bogs) vary typically between -80 to -50‰ and -25 to -0‰ [6] for $CH_4$ and $CO_2$, respectively (e.g. Whiticar et al., 1986; Hornibrook et al., 1997; Chasar et al., 2000; Hornibrook et al., 2000; Beer et al., 2008; Steinmann et al., 2008; Corbett et al., 2013). Depending on the available

Number: 1  Author:   Subject: Highlight  Date: 2018-09-21 11:58:34

Please, check whether this value is still relevant. There is no recent reference cited.

Number: 2  Author:   Subject: Replace  Date: 2018-09-21 11:56:07

28-fold

Number: 3  Author:   Subject: Highlight  Date: 2018-09-21 12:04:39

What about low temperatures? I am not sure bogs do exist in tropics (excluding mountain regions).

Number: 4  Author:   Subject: Highlight  Date: 2018-09-21 12:12:15

This is not clear: is this prerequisite for the oxidation? Not trapped=not oxidized? If so, how exactly could CH4 be trapped and how does CH4 oxidation occur in e.g. rice paddies?

Number: 5  Author:   Subject: Insert Text  Date: 2018-09-21 12:14:51

(delta 13C)

Number: 6  Author:   Subject: Highlight  Date: 2018-09-21 12:18:03

Actually, values could go even to positive range.

[revised manuscript text omitted]

**Number: 1  Author:   Subject: Highlight  Date: 2018-09-21 12:27:52**

Not clear. Do authors mean, reflect the signal of methanogenesis type? Please, rephrase.

**Number: 2  Author:   Subject: Highlight  Date: 2018-09-21 12:34:42**

How is this known? From the introduction above, it is not clear the pools are without (vasular) vegetation. Northern pools often contain aerenchimatous plants of different species compared to lawns and hummocks, or at least Sphagnum. Please, clarify above.

**Number: 3  Author:   Subject: Note  Date: 2018-09-21 12:41:34**

I find the Introduction a bit too extended, especially regarding the common knowledge about methane in the very beginning and peatlands in general. Authors could immediately start the story of the importance of southern peatlands and have the necessary information on peatlands' biochemistry and vegetation specialty in there. Then the information on the isotope issue would be sufficient to formulate hypotheses without any loss of logic.

exposed proximity to the Beagle channel with harsh winds (Grootjans et al., 2014) resulting in an oceanic climate with average daily temperatures of 6°C and annual precipitation of 500 mm (Fritz, 2012).

Large areas of the cushion bogs in Moat are covered by the cushion-forming plants *Astelia pumila* and *Donatia fascicularis*. The vegetation of the studied bog was composed of a mosaic of different plant communities characterized by specific plant
species with lawns of *A. pumila* being the prevailing microform. Sparsely vegetated pools of different size ($< 0.5$ m² - ~10 m²) and ~0.5 m depth were embedded in these *Astelia* lawns. *Sphagnum magellanicum* or *D. fascicularis* grew in small lawns (patches of few square meters) in close vicinity to the pools which we assume to provide protection against desiccation for *S. magellanicum*. *Tetroncium magellanicum*, a rush-like herb that does not form cushions but presumably establishes aerenchymous roots as it tolerates inundation (von Mehring, 2013), was associated predominantly with *Sphagnum* lawns but
also with *Astelia* or *Donatia* lawns. Peat formation started ~11000 years ago (Borromei et al., 2014), and previously, the bog was dominated by *S. magellanicum* while *A. pumila* invaded the area around 2600 years before present as determined by pollen analyses (Heusser, 1995). The peat below *Astelia* lawns was densely rooted with on the average 2.15 g DW $L^{-1}$ down to 1.7 m (Fritz et al., 2011). Mean root lifetimes of *A. pumila* has been estimated to be ~3-4 years indicating a high root turnover (Knorr et al., 2015).

**2.2 Sampling and analysis of solid peat and root biomass**

Peat coring in *Astelia* and *Sphagnum* lawns (data not shown) was done using a Russian peat corer (Eijkelkamp Agrisearch Equipment, Giesbeek, the Netherlands) down to a maximum depth of 7.5 m in *Astelia* lawns. Roots of *A. pumila* in *Astelia* lawn cores were sorted out and treated separately. The density of *D. fascicularis* root biomass was determined by cutting three 0.1 x 0.1 m sods down to a depth of 0.4 m. Peat and root biomass samples were oven dried at 70°C until constant weight to
calculate peat bulk density and porosity as well as *D. fascicularis* root density. Total C and N contents together with abundance of $^{15}N$ and $^{13}C$ in the solid peat were determined using an elemental analyser (EA 3000, EuroVector, Redavalle, Italy) connected to an isotope ratio mass spectrometer (IRMS, NU instruments, Wrexham, UK).

**2.3 Environmental variables**

Environmental variables were determined during two measurement campaigns in 2015 and 2016 at half-hourly intervals.
Photosynthetic active radiation (PAR, HOBO S-LIA-M003, Onset, USA, up to 2500 µmol m$^{-2}$ s$^{-1}$) and air temperature (HOBO S-TMB-M0x, Onset, USA) were recorded by a weather station (HOBO U30 NRC, Onset, USA) in both years during austral summer months. Water table fluctuations at two replicate sites in *Astelia* lawns (Levelogger Edge 3001 and Barologger Edge, Solinst, Canada; both installed in perforated PVC tubes) and soil temperature in four depths of 0.05, 0.1, 0.3 and 0.5 m (HOBO TMCx-HD and HOBO U-12-008, Onset, USA) were measured continuously.

Number: 1  Author:  Subject: Highlight  Date: 2018-09-21 12:55:33
Not any more? Should it be in Present Tense as the previous sentence?

Number: 2  Author:  Subject: Highlight  Date: 2018-09-21 12:58:22
Does this mean pools were nevertheless vegetated? If so, which species dominated? What was the bottom of such pools?

Number: 3  Author:  Subject: Highlight  Date: 2018-09-21 12:55:34

Number: 4  Author:  Subject: Highlight  Date: 2018-09-21 15:58:24
Liter of what, peat? For peat, could you provide other volumetric dimensions, e.g. dm-3?

[revised manuscript text omitted]

As indicated by high standard deviations, CH₄ emissions between replicates of all microforms were not consistent. Field observations suggested that CH₄ emissions of a microform were not only controlled by the predominant plant species, but might have been associated with additional features of the respective microform. Therefore additional microform features (i.e. number of *T. magellanicum* shoots, cover of *D. fascicularis*, extend of the microform in which a specific collar was installed) were assessed. However, as we did not expect such small-scale spatial variability between emissions of individual collars, the results of this survey given in Table 2 are of rather explorative, preliminary character as the low number of replicates (3 to 5 collars) in combination with several outliers barely allows any sound statistical analysis. Nevertheless, collars with elevated or comparatively high emissions had some features in common: Emissions from one *Astelia* lawn were with ~0.3 mmol CH₄ m$^{-2}$ d$^{-1}$ (significantly) higher in both sampling years compared to fluxes measured from the other *Astelia* lawns. This *Astelia* lawn collar with elevated emissions was characterized by (I) the presence of many *T. magellanicum* shoots together with a high share of *D. fascicularis* and (II) placed in a lawn with small extent surrounded by a mosaic of small pools and *D. fascicularis* patches (both smaller than 1 m$^2$). Surprisingly, also one *Donatia* lawn was a substantial CH₄ source of $2.10 \pm 0.14$ mmol m$^{-2}$ d$^{-1}$, even exceeding the highest emissions observed from *Sphagnum* lawns on all measurement occasions in 2016. Similar to the *Astelia* lawn with elevated emissions, the collars of *Donatia* and *Sphagnum* lawns with high emissions were characterized by (I) a high amount of *T. magellanicum* shoots and (II) placed in lawns surrounded by small pools and other patches of *D. fascicularis* with no *A. pumila* nearby. Emissions obtained from one *Sphagnum* lawn were with less than 0.3 mmol CH₄ m$^{-2}$ d$^{-1}$ in both sampling years significantly lower compared to emissions from other *Sphagnum* lawns. Vice versa,

Number: 1  Author:   Subject: Strikeout  Date: 2018-09-21 17:26:42

Number: 2  Author:   Subject: Highlight  Date: 2018-09-21 17:26:31

This is confusing: zero flux is not detectable (otherwise it is a positive or negative). Please, rephrase.
I am still wondering if outside air temperature is -0.5 C, how could 3 min be enough to measure any CH4 flux.

this collar with low *Sphagnum* emissions was characterized by (I) the lowest number of *T. magellanicum* shoots among all *Sphagnum* lawn collars and (II) installed in a small *Sphagnum* lawn surrounded by *A. pumila*.

**3.4 Pore water CH$_4$ and DIC concentration profiles**

Pore water profiles in peat columns below *Astelia* lawns and pools showed similar trends in pore water concentrations and during both sampling years. CH$_4$ concentrations were almost zero in the upper pore water profile below *Astelia* lawns (< 0.003 mmol L$^{-1}$) down to a depth of around 1.5 m, which corresponds to the zone where the rhizosphere was most pronounced (Fig. 4a). Below this depth, CH$_4$ concentrations sharply increased up to $0.25 \pm 0.08$ mmol L$^{-1}$ at 3 m depth. CH$_4$ concentrations in pore water profiles below pools resembled profiles obtained under *Astelia* lawns on elevated levels. Throughout the upper profile down to 1.5 m, CH$_4$ concentrations were ~0.03 mmol L$^{-1}$ with a peak around 0.3 m depth ($0.06 \pm 0.06$ mmol L$^{-1}$) and increased steeply to $0.40 \pm 0.25$ mmol L$^{-1}$ in 3 m depth. Maximum concentrations in comparable depths reached similar levels in both sampling years below *Astelia* lawns, but maximum concentrations below pools were about 3times higher in 2015 compared to 2016.

DIC predominantly occurred as dissolved CO$_2$ because of the low pH ranging from 3.36 to 4.77. In contrast to CH$_4$, DIC concentrations increased constantly with depth from around 1 mmol L$^{-1}$ near the surface to $2.60 \pm 1.0$ mmol L$^{-1}$ in 2 m depth below *Astelia* lawns and $2.94 \pm 1.1$ mmol L$^{-1}$ in 3 m depth below pools (Fig. 4b). With depth, DIC converged with CH$_4$ concentrations, which was reflected by DIC:CH$_4$ ratios that were extremely high in the rhizosphere below *Astelia* lawns exceeding 100 (Fig. 4c). Beneath the rhizosphere, ratios steeply approached values below 40. Under pools, ratios slightly decreased with depth but were mostly around 40 down to 1.5 m with two distinct peaks of very little CH$_4$ at the surface and around 0.8 m depth. In deep peat layers below the rhizosphere, DIC:CH$_4$ ratios detected under *Astelia* lawns and pools converged and reached lowest ratios of ~10 and ~5, respectively.

Visual inspection of concentration profiles but also modelling of CH$_4$ production and consumption rates indicated a predominance of CH$_4$ consumption throughout the whole rhizosphere below *Astelia* lawns and in roughly corresponding depths under pools (Fig. 4a, Fig. 6). Maximum CH$_4$ consumption was identified in the lower rhizosphere around 1.5-2 m where the increase in CH$_4$ concentration was most pronounced. Deep peat layers below the rhizosphere of *Astelia* lawns and similar depths below pools were considered as CH$_4$ sources. In these depths, the high concentration levels of CH$_4$ (up to $0.40 \pm 0.25$ mmol L$^{-1}$) sustain substantial upward diffusion of CH$_4$ following the concentration gradient into the consumption zone within 1 the rhizosphere.

**3.5 Composition of pore water in carbon isotopic values 2nd apparent fractionation**

Values of δ$^{13}$C in CH$_4$ did not show a clear depth trend below *Astelia* lawns and scattered in the upper profile down to 2 m depth between $-87.2 \pm 10.1$ to $-72.1 \pm 10.3$‰ (Fig. 4d). Isotopic δ$^{13}$C-CH$_4$ values below pools became less negative with depth

Number: 1  Author:   Subject: Highlight  Date: 2018-09-21 17:53:34

There was no rhizosphere below pools, so what then caused the gradient?

Number: 2  Author:   Subject: Highlight  Date: 2018-09-21 17:54:58

[revised manuscript text omitted]

**Number: 1  Author:  Subject: Highlight  Date: 2018-09-24 12:42:21**

With this, authors attempt to oversell their results: "scattered between" seemingly indicate no significant difference. Indeed, Fig. 4d demonstrate rather narrow d13C-CH4 range along the whole depth profile. So, in fact, d13C-CH4 signal alone was not informative enough to approve the strong oxidative properties of rhizosphere of A. pumila. I agree that both methanogenesis and oxidation may co-exist in close vicinity, but still it may not explain lack of d13C-CH4 variation between upper and lower horizons unless CH4 produced in the rhizosphere region is even more depleted in 13C than in deeper layers. The explanation of this phenomenon because of "more reduced...microsites" is not fully clear. More than below the rhizosphere? Why?

**Number: 2  Author:  Subject: Note  Date: 2018-09-24 12:54:14**

According to this oxidation concept, the most 13C enriched CH4 has to be allocated at the shallowest depth. However, in contrast, it is ca. 10‰ more depleted than next depth levels (20-50 cm). In addition, d13C-CO2 is relatively more enriched that in deeper layers. How is this possible?

**Number: 3  Author:  Subject: Highlight  Date: 2018-09-24 12:58:19**

This information is already repetition of the message above. I suggest to merge both parts telling the story as here but with the reference to results as in the previous paragraph. Otherwise, it is excessive.

**Number: 4  Author:  Subject: Highlight  Date: 2018-09-24 13:07:25**

This contradicts to the data measured: having acetoclastic methanogenesis and co-existence of oxidation should generate much more enriched d13C-CH4 values in comparison to deep peat. Fig. 4d cannot support this. Seemingly, change of fractionation factor with depth was not significant either. The available data are not enough to approve existence of acetocalstic methaogenesis. Please, discuss this.

**Number: 5  Author:  Subject: Highlight  Date: 2018-09-24 13:09:12**

Again, there is not enough evidence to support the hypothesis. As it is stated, this is speculation and should be rephrased.

**Number: 6  Author:  Subject: Highlight  Date: 2018-09-24 13:10:55**

Yes, but it was small below the rhizosphere too! Speculation!

**Number: 7  Author:  Subject: Highlight  Date: 2018-09-24 13:12:48**

They also increased at the very top of profile. No information about significance of differences in fractionation factor between depths is provided.

supposed to be substantial, $O_2$ release by roots still exceeded consumption, as demonstrated by $O_2$ saturation reaching up to 10%. This $O_2$ supply may thus not only fuel $CH_4$ oxidation but also heterotrophic respiration using molecular oxygen or other electron acceptors regenerated by this $O_2$ input (Colmer, 2003; Mainiero and Kazda; 2005). Besides aerobic respiration in oxidized microsites attached to roots, other likely sources of $CO_2$ could have been dissimilatory sulfate reduction in anaerobic microsites of surface peat layers down to 0.4 m with sulfate originating e.g. from sea spray (Kleinebecker et al., 2008; Broder et al., 2015). Elevated $H_2$ levels in the rhizosphere indicated high $H_2$ production by fermentation compared to consumption by e.g. sulfate reduction, presumably due to high root litter input and root exudates (Knorr et al., 2009; Estop-Aragonés et al., 2013).

**4.3 Pools – low emission scenario**

Pools exhibited with $0.23 \pm 0.25$ mmol $CH_4$ m$^{-2}$ d$^{-1}$ second lowest emissions of the microforms under study (Fig. 3). Such low emissions measured from pools were somewhat surprising since they have been previously reported to be similar to *Sphagnum* lawns (Fritz et al., 2011) and were expected to be relevant $CH_4$ sources because of prevailing anoxic conditions (Blodau, 2002). Moreover, other studies identified pools as hot-spots of $CH_4$ emissions from peatlands (Hamilton et al., 1994; Pelletier et al., 2014; Burger et al., 2016). This was clearly not the case in our study, as emissions were in the same order of magnitude as pool fluxes in South Patagonian *Sphagnum* bogs (Lehmann et al., 2016).

Low emissions from pools were associated with $CH_4$ pore water concentrations following those below *Astelia* lawns on slightly elevated levels (Fig. 4a). These results verified our second hypothesis stating that pore water $CH_4$ concentrations of non-rooted[2] peat below pools remains less affected by $CH_4$ oxidation compared to *Astelia* lawns. Nevertheless, it was surprising that $CH_4$ concentration profiles below pools clearly resembled profiles obtained under *Astelia* lawns. Such concentration profile[3]

suggests that *A. pumila* roots controlled even $CH_4$ dynamics below adjacent microforms as visualized on a conceptual level in Figure 6. To explain low $CH_4$ concentrations in upper peat layers below pools, we suggest that the thorough $CH_4$ oxidation below *Astelia* lawns can also establish a lateral concentration gradient[4] Thereby, $CH_4$ from adjacent microforms gets redistributed to the rhizosphere of *Astelia* lawns where it is consumed. Throughout the rhizosphere of *Astelia* lawns and in[5] corresponding depths below pools, we observed a highly decomposed and amorphous peat as previously described for cushion bogs (Ruthsatz and Villagran, 1991; Fritz et al., 2011) indicating a very high water permeability that would together with a high water table position enable such lateral redistribution (Baird et al., 2016). The assumption of lateral pore water movement was supported by a peak in DIC:$CH_4$ ratios accompanied by very low $CH_4$ concentrations around 0.8 m below pools (Fig. 4c) as upward diffusion against the concentration gradient is impossible. Lateral root ingrowth and $O_2$ supply from adjacent *Astelia*[6] lawns could furthermore keep $CH_4$ concentrations in upper peat layers below pools at a low level, in particular if pools are small.

As observed under *Astelia* lawns, $CH_4$ was produced in deep peat layers below pools also predominantly by hydrogenotrophic methanogenesis and diffused upwards. In upper peat layers influenced by lateral movement of pore water, oxidation of $CH_4$

**Number: 1  Author:  Subject: Highlight  Date: 2018-09-24 13:38:45**

This may also mean occurrence of hydrogenotrophic methanogenesis in anaerobic rhizosphere zones. For example, Galand et al. (2002) FEMS, demonstrated dominance of H2-trophic methanogens in upper peat layer in a boreal northern peatland. Is there any evidence for southern peatlands too? This may partly explain relatively depleted d13C values of CH4 in the rhizosphere zone.

**Number: 2  Author:  Subject: Highlight  Date: 2018-09-24 14:44:12**

This can be misleading: 2nd hypothesis specified processes based on isotopic values, whereas here authors refer more to the concentrations/fluxes thereby considering rather 1st hypothesis. The latter, however was not supported. Please, rephrase.
Also regarding the 2nd hypothesis, "less affected" is not appropriate for the hypothesis. Please, check the respective comment above.

**Number: 3  Author:  Subject: Highlight  Date: 2018-09-24 14:48:30**

This is not clear: How could roots of A. pumila appear under pools? Was this observed during coring? If so, then the conceptual diagram should demonstrate that roots of A. pumila expand below pools. Check and correct accordingly.

**Number: 4  Author:  Subject: Highlight  Date: 2018-09-24 14:51:27**

Of what, CH4 or oxygen?

**Number: 5  Author:  Subject: Highlight  Date: 2018-09-24 15:11:08**

This statement is unclear: whereas gas diffusion along gradient is clear for me (from pools to lawns) water movement is not the same. Pools are local depressions, so water should flow from lawns into pools. If this flow is so low, then the gas diffusion in opposite direction can be stronger, but this means almost standing water. In case there is a lateral flow of water (what is very natural), then the gas flow can't be counter to it. Therefore, I could understand the inflow of oxygen from lawns into pools, but not CH4 from pools to lawns. The overall picture may change if there is a slope, but then lawns and pools have to be arranged accordingly. Pools will get matter of those lawns which are elevated and transfer it downwards to other lawns. If there is a slope on the site, then the conceptual figure should somehow reflect it. Check!

**Number: 6  Author:  Subject: Highlight  Date: 2018-09-24 15:16:30**

What is meant, suppression of methanogenesis or CH4 oxidation? This is important in context of measured isotope values. Please, specify.

[revised manuscript text omitted]

*Acknowledgement.* This study was carried out within the research project CANDYbog which was funded by the Deutsche Forschungsgemeinschaft (German Research Foundation, DFG; Grant No. KL2265/3-1 and BL 563/19-1). We highly appreciated the Centro Austral de Investigaciones Científicas (CADIC-CONICET) providing facilities that enabled our fieldwork in Ushuaia. The Prefectura Naval Argentina is acknowledged for offering facilities supporting our fieldwork. Lucas Varela of the La Posta Hostel family is appreciated for offering working space and logistics. We wish to acknowledge the laborious field assistance of Isabella Närdemann, Carla Bockermann, Claudia Frank, Bettina Breuer, Lina Birkner, Juliane Kohlstruck, Laura Jansen, David Holl, Lars Kutzbach, María Florencia Castagnani, María Noel Szudruk Pascual. All analyses of this study were carried out in the laboratory of the institute of Landscape Ecology. The assistance of Ulrike Berning-Mader, Sebastian R. Schmidt, Ronya Wallis, Sabrina Knaack, Sina Berger, Simona Bonaiuti and Leandra Praetzel is greatly acknowledged. We thank Peter Sulmann for technical assistance.

No Comments.

**References**

Agethen, S., Sander, M., Waldemer, C., Knorr, K.-H.: Only low methane emissions from restored cutover bogs—The impact of vascular plants. In preparation, 2017.

[revised manuscript text omitted]

No Comments.

---

## Author Comment (AC1) · 30 Oct 2018

Dear reviewer,

we appreciate your constructive comments that will help to improve our manuscript. Please find our response (blue) to your comments (black) below.

**Reviewer 1:**

General comments

This is a carefully done study about the production, oxidation and emission of CH4 in Patagonian bog, the results are of considerable interest and the paper is well written. However, some points need clarifying and certain statements require further justification.

Specific comments

1. the authors should not ignore that acetogenesis might be important in anaerobic environments when H2 partial pressures are high and temperatures are low. Acetogens can outcompete methanogens at low temperature, as many acetogens seem to have a higher growth rate at low temperature than most methanogens (Kotsyurbenko et al., 1996, 2001). If acetogenesis process is active in the bog, the δ3C value of actetate in the porewater will be largely decreased because of the substantial fractionation during acetate production from CO2 and H2. And resultantly, the 13C value of CH4 will also be lower and resulted in larger apparent isotopic fractionation factor (ac) between CO2 and CH4. Therefore, it's difficult to determine the relative importance of acetoclastic versus hydrogenotrophic methanogenesis pathway without the 13C value of acetate in this study.

*We agree with the reviewer, that the d13C-CH4 signal was surprisingly low and therefore needs an explanation. Indeed, a strong effect of only methanotrophy should result in a less negative signature of d13C in CH4. We attempt to explain the low d13C-CH4 signal by the occurrence of microsite that create a mixed isotopic signal from methane production and consumption. As the reviewer correctly points out, the occurrence of acetogenesis is a reasonable explanation for the surprisingly negative d13C-CH4 signatures. Reviewer 2 suggests occurrence of hydrogenotrophic methanogenesis as a possible explanation. We agree that both pathways could explain the pattern in the d13C-CH4 signal, although we propose in accordance with the reviewer that hydrogenotrophic methanogenesis was probably relatively more important below pools while acetoclastic methanogenesis in combination with acetogenesis seemed to contribute more below Astelia lawns. But as we did not quantify other parameters such as labile organic matter from roots, acetate concentrations or its carbon isotopic signatures, we cannot clearly determine the relative importance of both pathways for our study, as the reviewer correctly states.*
*In the revised discussion, we will emphasize that the depleted d13C-CH4 signal is an unexpected result that needs better explanation. We will better explain possible sources for depleted 13C-CH4 within the rhizosphere and discuss the possibility of the occurrence of acetogenesis. The difficulties to separate the isotopic effects arising from methanogenic pathways will be elaborated. We will try to balance the suggestions of both reviewers by discussing the arguments for both, hydrogenotrophic and acetoclastic methanogenesis and acetogenesis, carefully.*

2. In the first page, line 26-28, it's stated that: "Below the rhizosphere. . .. . .CH4 was predominantly produced by hydrogenotrophic methanogenesis". In fact, data in Figure 4def showed that the hydrogenotrophic pathway had higher contribution to CH4 in the pool, while the acetoclastic pathway must play relatively more important role for the CH4 production below the rhizosphere of Astelia Lawn. This is consistent with the supply of labile organic carbon from the root exudates of Astelia. To sum up, I think it's difficult to conclude that CH4 is mainly produced from the hydrogenotrophic pathway below the rhizosphere of Astelia.

*We agree with the reviewer that hydrogenotrophic methanogenesis was probably relatively more important below pools while acetoclastic methanogenesis seemed to contribute more below Astelia lawns. This seemed to have been even true below the rhizosphere of Astelia lawns, but is a result hard to explain from the data obtained in our study. As the reviewer correctly points out, labile organic matter from roots could be a possible explanation. But as we did not quantify other parameters such as labile organic matter from roots, acetate concentrations or its carbon isotopic signatures, we cannot clearly determine the relative importance of both pathways for our study, as the reviewer correctly states in the general comment 1.*

*In the revised abstract, we will rephrase that sentence on page 1, line 26-28. It will be emphasized in the discussion that the depleted d13C-CH4 signal is an unexpected result which needs better explanation. We will better explain possible sources for depleted 13C-CH4 and discuss the possibility of the occurrence of acetogenesis. The difficulties to separate the isotopic effects arising from methanogenic pathways will be elaborated. We will try to balance the suggestions of both reviewers by discussing the arguments for both, hydrogenotrophic and acetoclastic methanogenesis and acetogenesis, carefully.*

3. It's stated that mean root lifetimes of A. pumila has been estimated to be ~3-4 years. So, whether the production and oxidation of CH4 will be strongly affected in case of the turnover of large amounts of roots?

*It is an interesting point that turnover i.e. presence and activity of roots should largely determine the occurrence of microsites. One could expect that temporal and spatial expansion of microsites is not static but varies with root life time and turnover. As reviewer 2 points to a partly speculative discussion about microsites in the current version of the manuscript, we will briefly include this aspect into the revised discussion.*

4. Please check Table 3, the data in the last three columns are in wrong places.
*Yes, the reviewer is correct. We will correct Table 3 accordingly.*

---

## Author Comment (AC2) · 30 Oct 2018

Dear reviewer,

we appreciate your thoughtful and detailed review! Your constructive comments will help to improve our manuscript and particularly the discussion. Please find our response (blue) to your comments (black) below.

**Reviewer 2:**

General comments

The MS of Münchberger and co-authors contributes to the knowledge on processes of methane turnover (production+oxidation) and transport in rarely studied southern bogs. Authors combined field sampling with the advanced analytics (porewater chemistry and stable isotope analyses) to report relationships and peculiar mechanisms between microrelief forms, dominating vegetation communities and the net processes affecting the CH4 efflux from the Patagonian peatland during two consecutive summer seasons. Field-based studies are critically important for understanding processes related to functioning of ecosystems and therefore interesting for the broad scientific community. Accepting the field experiments typically operate with much larger spatial and temporal variability in measured parameters (and as the result, relatively lower statistical power as compared to controlled conditions), still there are several issues which I would like to point out for the discussion and improvement. Below authors find general comments while specific recommendations and technical corrections are incorporated directly in the draft file attached.

1. First of all, the MS is rather long and too repetitive and descriptive. Thus, the Introduction is definitely too extended, especially regarding the common knowledge about methane in the very beginning and peatlands in general. Authors could immediately start the story of the importance of southern peatlands and have the necessary information on peatlands' biochemistry and vegetation specialty in there. Then the information on the isotope issue would be sufficient to formulate hypotheses without any loss of logic.

*We understand your concern and appreciate your suggestions of how the topic of the manuscript could be introduced in a more straightforward way. However, the strength of our dataset is the combination of both, a comprehensive chamber measurement campaign and advanced pore water chemistry. To address our manuscript to the readers of both communities, researcher focusing on gas exchange in peatlands as well as those mainly dealing with biogeochemical processes in the peat, we decided to shortly introduce the main mechanisms of both research areas. Southern peatlands are introduced only towards the end of the introduction since the rhizosphere effects on methane dynamics due to aerenchymatic vascular plants are of relevance for other peatlands with a dense cover of vascular plants such as rushes and sedges. We believe that starting the introduction with the importance of southern peatlands would result in a too narrowed topic of the manuscript. Therefore, and as reviewer 1 did not comment on the structure and length of the introduction, we hope that reviewer 2 could agree with our positions here.*
*Nevertheless, we will try to shorten descriptive as well as introducing and transitional passages in our manuscript. As we understand that parts of the discussion are too repetitive and descriptive, we will also shorten the respective paragraphs. We kindly refer to our answer to general comment 4 to 7 below for details on this aspect.*

2. In the proposed hypotheses, it has to be clearer why pools are so much different from lawns in terms of methanogenesis pathways. This was not strait forward from the introduction; I suggest to omit statements as "remains less affected" because they are more confusing then explanatory; please, rephrase.

*We agree that the vegetation in the pools needs more explanation in the introduction as otherwise the development of the hypothesis III is not straightforward. The hypothesis will*

*be rephrased accordingly. Also hypothesis II will be rephrased into a better explanatory version.*

3. In the Methods section, I was confused with relatively short time (3 min) of chamber exposition even under the conditions of rather low atmospheric temperatures and low fluxes expected. Why also transparent and not opaque chambers were used for CH4 fluxes measurement?

*Indeed, chamber measurements with a chamber not connected to a fast gas analyzer need up to 30 minutes or more to determine methane fluxes. The chamber used in our study was connected to a portable gas analyzers (Ultraportable Greenhouse Gas Analyzer, 915-001, Los Gatos Research) with a 1 hz sampling rate. The instrument accuracy according to the manufacturer was < 2 ppb. Therefore, this instrument provides the opportunity to determine concentration changes even at low CH4 concentrations and within a short period of time (see for example McEwin et al., 2015; Berger et al., 2018 or Mastepanov et al., 2013 who used a similar gas analyzer with 1 hz sampling rate). Test measurements with a prolonged closure time and instantaneous on-site monitoring of gas concentrations changes within the chamber proved that a short measurement time was sufficient to determine the CH4 fluxes also at our study site. From this, we could furthermore exclude that zero fluxes are a methodological artifact.*

*And indeed, despite the short measurement time and low fluxes, only 105 of 537 flux measurements showed a concentration increase with a slope not significantly different from zero (page 7, lines 18-19). During all other measurements (at least a small but) significant concentration change was observed already during the short measurement time.*

*Low or zero fluxes were even not an artifact due to a short measurement time at low temperatures. Otherwise collars with low or zero fluxes would have shown a response to temperature. We kindly refer to figure S02 in the supplement provided to the manuscript. This figure show that most individual collars did not show any response to temperature. Please compare also to the explanation on page 7, lines 22-26.*

*We used transparent and opaque chambers since the Los Gatos gas analyzer can measure CH4 and CO2 simultaneously. Thus, our measurement campaign was designed to determine also the NEE. As the CH4 fluxes did not differ systematically between light and dark measurements, we included also the light measurements in our data set to increase the sample size.*

4. Discussion section contains repetitive and partly speculative information and therefore is currently too long. For instance, in the discussion of results on 13C-CH4 depth profile (page 14, lines 5-8) authors seemingly "oversell" their results: "scattered between" may also indicate no significant difference (this is not clear from the data). Indeed, Fig. 4d demonstrates rather narrow d13C-CH4 range along the whole depth profile. So, in fact, d13C-CH4 signal alone was not informative enough to approve the strong oxidative properties of rhizosphere of A. pumila. I agree that both methanogenesis and oxidation may co-exist in close vicinity, but still it may not explain lack of d13C-CH4 variation between upper and lower horizons unless CH4 produced in the rhizosphere region is even more depleted in 13C than in deeper layers. The explanation of this phenomenon because of "more reduced...microsites" is not fully clear. More than below the rhizosphere? Why?

*We understand that parts of the discussion are too repetitive and partly speculative. The comparatively small shift of the mean d13C-CH4 signature to more enriched values within the rhizosphere was a surprising and unexpected result also to us. Indeed, a strong effect of only methanotrophy should result in a less negative signature of d13C in CH4 compared to the signature below the rhizosphere. Therefore we completely agree with the reviewer, that the d13C-CH4 signal alone does not provide a clear indication for oxidative effects in the rhizosphere. Taking into account the near-zero CH4 emissions, high DIC:CH4 ratios and a d13C-CH4 depth pattern not following the d13C-CO2 and a depleted d13C-CO2 signal, we can nevertheless only explain these results by a strong effect of methanotrophy.*

*In addition, we attempted to explain the lack of variation in the mean d13C-CH4 signal. Throughout the rhizosphere, the d13C-CH4 signal was associated with a wider standard variation compared to deeper peat layers. Our possible explanation for this is that the mean d13C-CH4 signal represents a mixed signal from methane production and consumption and, thus, indicates a co-existence of aerobic and anaerobic microsites. Maybe our explanations were too detailed here and thus became speculative. In fact, the word scatter was indeed used incorrectly here to describe the wider standard variation.*

*As the reviewer correctly points out, the occurrence of hydrogenotrophic methanogenesis is a possible explanation for surprisingly negative d13C-CH4 signatures. Reviewer 1 suggests occurrence of acetogenesis as a possible explanation. We agree that both pathways could explain the pattern in the d13C-CH4 signal, although we propose in accordance with reviewer 1 that hydrogenotrophic methanogenesis was probably relatively more important below pools while acetoclastic methanogenesis in combination with acetogenesis seemed to contribute more below Astelia lawns. But as we did not quantify other parameters such as labile organic matter from roots, acetate concentrations or its carbon isotopic signatures, we cannot clearly determine the relative importance of both pathways for our study. We kindly refer to our answer to general comment 5 below for more details.*

*In the revised discussion, the word "scatter will be rephrased. We will follow the helpful suggestion of the reviewer (page 14, comment 3) and incorporate aspects of the paragraph on page 14, lines 17-32 into the paragraph above (lines 7 and following). Thereby, repetitive and speculative information concerning the coexistence of microsites will be reduced. We will furthermore emphasize that the lack of d13C-CH4 variation between upper and lower horizons is an unexpected result. We agree with the reviewer that the d13C-CH4 depths pattern needs further explanation that is so far missing in the discussion. We will try to balance the suggestions of both reviewers by discussing the arguments for both, hydrogenotrophic and acetoclastic methanogenesis and acetogenesis, carefully. The sentence "more reduced... microsites" (p 14, lines 7-8) does not refer to "more reduced" compared to below the rhizosphere, but compared to oxidized microsites in close vicinity to reduced microsites. This misleading phrase will be clarified.*

5. Contribution of acetoclastic pathway to methanogenesis in the rhizosphere of A. pumila was not convincingly verified (e.g. page 14, lines 23-27) and looks therefore speculative: having acetoclastic methanogenesis and co-existence of oxidation should generate much more enriched d13C-CH4 values in comparison to deep peat. Fig. 4d cannot support this. Seemingly, change of fractionation factor with depth was not significant either. The available data are not enough to approve existence of acetoclastic methanogenesis, and this has to be acknowledged.

*We agree with the reviewer, that a clear effect of methanotrophy and acetoclastic methanogenesis on the d13C-CH4 signature should result in an enriched d13C-CH4 signal and a distinctly lower fractionation factor throughout the rhizosphere compared to deeper peat layers. We were thus surprised by the small (and probably not statistically significant) change in the fractionation factor. Thus, we attempt to explain a fractionation factor in the overlap range of ac from hydrogenotrophic and acetoclastic methanogenesis.*

*As already observed for the d13C-CH4 signal, also the fractionation factor shows a wider standard variation within the rhizosphere. The standard variation of the fractionation factor even tended to be larger with increasing depth down to the lower boundary of the rhizosphere. This pattern comes along with a presumably with depth decreasing root density (Fritz et al., 2011) in these depths. We therefore interpret this as a further indication for the occurrence of microsites as lower root density makes a more heterogeneous peat matrix more likely and thus a higher variation in the fractionation factor between sampling sites.*

*We agree with the specific recommendation on page 15, comment 1 that occurrence of hydrogenotrophic methanogenesis at elevated H2 levels in surface peat layers might be a possible source for depleted 13C and an explanation for only small changes and a comparatively higher standard variation in the fractionation factor. Another reasonable explanation is the occurrence of acetogenesis, as suggested by reviewer 1.*

*In the revised discussion, it will be underlined that only small changes in the fractionation factor between upper and lower horizons are an unexpected result if methanotrophic effects are assumed. We will better explain possible sources for depleted 13C-CH4 within the rhizosphere and discuss the possibility of the occurrence of acetogenesis. The difficulties to separate the isotopic effects arising from methanogenic pathways will be elaborated. As we did not quantify other parameters such as labile organic matter from roots, acetate concentrations or its carbon isotopic signatures, we will carefully discuss the possibility of both, hydrogenotrophic and acetoclastic methanogenesis based on the data obtained in our study.*

6. Another critical point is again a speculative discussion of the results on pools and lateral flows on the site (page 15, lines 23-35). Explanations on gas diffusion along gradient were clear for me (from pools to lawns) but water movement is not the same. Pools are local depressions, so water should flow from lawns into pools. If this flow is so low, then the gas diffusion in opposite direction can be stronger, but this means almost standing water. In case there is a lateral flow of water (what is very natural), then the gas flow can't be counter to it. Therefore, I could understand the inflow of oxygen from lawns into pools, but not CH4 from pools to lawns. The overall picture may change if there is a slope, but then lawns and pools have to be arranged accordingly. Pools will get matter of those lawns which are exposed higher and transfer it downwards to other lawns. If there is a slope on the site, then the conceptual figure should somehow reflect it. Such important information was not provided in Mat&Meth or any other parts of MS.

*We thank the reviewer for this careful and critical examination of our concept. We attempt to explain low CH4 concentrations in the pore water below pools, as the isotopic signals below pools did not indicate a methanotrophic effect. Upward diffusion of CH4 against the concentration gradient is not possible and CH4 emissions were low, so we can only explain this by lateral exchange of CH4.*
*Indeed, pools are local depressions in the micro-relief, and we therefore agree that water should flow from lawns into pools. Nevertheless, the micro-relief is not very pronounced at our study site with Astelia lawns being elevated by only about 5-20 cm above the water table and the pool surface. This is indicated in the conceptual figure. We therefore assume, that the micro-relief does not exert much impact on the water flow in deeper peat layers and water flow from lawns to pools should be restricted to the uppermost decimeters of the peat profile. In contrast, the rhizosphere stretches over almost 2 m within highly decomposed peat. So we propose that there is a large zone with negligible water flow throughout the rhizosphere where water movement from pools to lawns would be reasonable. Due to low water movement, diffusive transport dominates and both, CH4 transport from pools to lawns and O2 transport from lawns to pools could be reasonable. We will explain this concept more clearly in the revised version of the discussion.*

7. The section 4.4. is rather long and at several places contains repetitive text (e.g. page 17, lines 15-17, 21-23, 27-28; the effect of A. pumila roots was very clear, no need to repeat many times). I recommend condensing text strongly.

*We understand that the interpretation of the observed emission pattern is repetitive concerning biogeochemical processes in the peat. As reviewer 1 did not comment on the length of this section, we will carefully shorten repetitive explanations in paragraph 4.4., starting in line 12 on page 17.*

8. Depending on the available information from authors, the conceptual Fig. 6 can be changed (see more detailed comments in the text).

*We kindly refer to our answer to general comment 6.*

Specific recommendations and technical corrections incorporated in the draft file of the manuscript

- Comment 1, line 25: How was this tested? If so, how far from root surface the "suppressive effect" is possible?

*We agree, this interpretation of our results written in the abstract needs more explanation. Will be rephrased.*
- Comment 2, lines 30-32: Please, rephrase the sentence in a more simplistic way. Too difficult to read and understand.

*Will be rephrased.*

- Comment 1, lines 6-7: Please, check whether this value is still relevant. There is no recent reference cited.

*Recent references will be checked.*
- Comment 2, line 7: 28-fold

*This technical correction will be changed accordingly.*
- Comment 3, lines 10-12: What about low temperatures? I am not sure bogs do exist in tropics (excluding mountain regions).

*We will add some information about the importance of temperature.*
- Comment 4, line 17: This is not clear: is this prerequisite for the oxidation? Not trapped=not oxidized? If so, how exactly could $CH_4$ be trapped and how does $CH_4$ oxidation occur in e.g. rice paddies?

*Indeed, this wording is confusing. We will rephrase the sentence.*
- Comment 5, line 30: (delta 13C)

*Will be added.*
- Comment 6, line 32: Actually, values could go even to positive range.

*Will be corrected.*

- Comment 1, line 17: Not clear. Do authors mean, reflect the signal of methanogenesis type? Please, rephrase.

*Indeed, the hypothesis is not clear. We will rephrase it.*
- Comment 2, lines 19-20: How is this known? From the introduction above, it is not clear the pools are without (vasular) vegetation. Northern pools often contain aerenchimatous plants of different species compared to lawns and hummocks, or at least Sphagnum. Please, clarify above.

*This comment was already answered above in the "general comments" section, comment 2.*
- Comment 3, line 23: I find the Introduction a bit too extended, especially regarding the common knowledge about methane in the very beginning and peatlands in general. Authors could immediately start the story of the importance of southern peatlands and have the necessary information on peatlands' biochemistry and vegetation specialty in there. Then the information on the isotope issue would be sufficient to formulate hypotheses without any loss of logic.

*This comment was already answered above in the "general comments" section, comment 1.*

- Comment 1, line 4: Not any more? Should it be in Present Tense as the previous sentence?

*The tense will be corrected and the whole manuscript checked for correct tense.*
- Comment 2, line 5: Does this mean pools were nevertheless vegetated? If so, which species dominated? What was the bottom of such pools?

*The vegetation in the pools will be described in more detail.*
- Comment 3, line 6:

*The tense will be corrected and the whole manuscript checked for correct tense.*
- Comment 4, line 7: Liter of what, peat? For peat, could you provide other volumetric dimensions, e.g. dm-3?

*Will be clarified and checked.*

- Comment 1, line 16: This could be too short time for CH4 flux measurement especially if outside temperatures were relatively low. How was it determined? Could zero fluxes be the reason of short exposure time?

*This comment was already answered above in the "general comments" section, comment 3.*

- Comment 1, line 4: was

*The tense will be corrected and the whole manuscript checked for correct tense.*
- Comment 2, line 24: How? What was the volume of the sample? 3 ml? For such small volumes a separate device (Small volume unit) is necessary. Please, expalin.

*Yes, the reviewer is correct. The missing information in the description of the device will be added.*
- Comment 3, line 30: Confusing: organic or inorganic? Please correct.

*Will be changed to "dissolved inorganic carbon".*

- Comment 1, line 22: Again, liter is not clear for peat as volume containing roots.

*Will be clarified as for comment 4 on page 5.*
- Comment 2, line 23: Please, provide here a value with the reference to study (studies). This will help better compare the differences between plant species.

*This information will be added.*

- Comment 1, line 1:

*This technical correction will be changed accordingly.*
- Comment 2, line 1: This is confusing: zero flux is not detectable (otherwise it is a positive or negative). Please, rephrase. I am still wondering if outside air temperature is -0.5 C, how could 3 min be enough to measure any CH4 flux.

*We agree that this phrase is confusing. We will rephrase it. Concerning a short measurement time even under low temperature, we refer to our answer to comment 3 in the "general comments" section.*

- Comment 1, lines 26-27: There was no rhizosphere below pools, so what then caused the gradient?

*We agree that the phrase is misleading. We will describe the concentration gradients and explain subsequent diffusion pathways in more detail here.*
- Comment 2, line 28: Suggest to rephrase: Carbon isotopic values in pore water and apparent fractionation.

*We agree and will rephrase the section title.*

- Comment 1, line 7: fractionation

*This technical correction will be changed accordingly.*
- Comment 2, line 22: with?

*This technical correction will be changed accordingly.*

− Comment 1, lines 30-31: Please, check the definitions: typically, "alternative" means alternative to oxygen. So, oxygen cannot be alternative to itself.

*We agree with the reviewer. The phrase was misleading and will be changed to "… either O2 or alternative electron acceptors…"*

− Comment 1, lines 6-7: With this, authors attempt to oversell their results: "scattered between" seemingly indicate no significant difference. Indeed, Fig. 4d demonstrate rather narrow d13C-CH4 range along the whole depth profile. So, in fact, d13C-CH4 signal alone was not informative enough to approve the strong oxidative properties of rhizosphere of A. pumila. I agree that both methanogenesis and oxidation may co-exist in close vicinity, but still it may not explain lack of d13C-CH4 variation between upper and lower horizons unless CH4 produced in the rhizosphere region is even more depleted in 13C than in deeper layers. The explanation of this phenomenon because of "more reduced...microsites" is not fully clear. More than below the rhizosphere? Why?

*This comment was already answered above in the "general comments" section, comment 4.*

− Comment 2, line 17: According to this oxidation concept, the most 13C enriched CH4 has to be allocated at the shallowest depth. However, in contrast, it is ca. 10‰ more depleted than next depth levels (20-50 cm). In addition, d13C-CO2 is relatively more enriched that in deeper layers. How is this possible?

*We inspected again our dataset to explain this. During a measurement, the isotopic signal of each sample is determined repetitively. So in fact, the signal determined from one sample is a mean of many measurements. To further improve the data quality, we excluded the isotopic signal of one sample with an elevated SD. This results now in a less enriched mean of d13C-CO2 in the uppermost peat layer.*
*The sampling devices were installed below the water table, but only mean of water table is given in the figures. We will check the line denoting the water table in the figures which is not exactly at the correct place. So, in the uppermost depth not much influence from roots can be expected. Accordingly, the CH4 was not enriched due to methanotrophic effects, but comparatively depleted by methanogenesis (please compare to answers to comment 4 and 5 in the general comment section).*

− Comment 3, lines 17-22: This information is already repetition of the message above. I suggest to merge both parts telling the story as here but with the reference to results as in the previous paragraph. Otherwise, it is excessive.

*We understand that this part of the discussion is repetitive and kindly refer to our answer to comment 4 in the "general comments" section. We will follow the helpful suggestion of the reviewer and incorporate aspects of the paragraph on page 14, lines 17-32 into the paragraph above (lines 7 and following).*

− Comment 4, lines 23-26: This contradicts to the data measured: having acetoclastic methanogenesis and co-existence of oxidation should generate much more enriched d13C-CH4 values in comparison to deep peat. Fig. 4d cannot support this. Seemingly, change of fractionation factor with depth was not significant either. The available data are not enough to approve existence of acetoclastic methaogenesis. Please, discuss this

*This comment was already answered in the "general comments" section, comment 4 and 5.*

− Comment 5, lines 27-28: Again, there is not enough evidence to support the hypothesis. As it is stated, this is speculation and should be rephrased.

*We will phrase this information more carefully and kindly refer to our answer to general comment 5 in the "general comments" section.*

− Comment 6, lines 29-30: Yes, but it was small below the rhizosphere too! Speculation!

*This comment was already answered in the "general comments" section, comment 4 and 5.*

− Comment 7, line 32: They also increased at the very top of profile. No information about significance of differences in fractionation factor between depths is provided.

*This comment was already answered in the "general comments" section, comment 5.*

- Comment 1, line 6: This may also mean occurrence of hydrogenotrophic methanogenesis in anaerobic rhizosphere zones. For example, Galand et al. (2002) FEMS, demonstrated dominance of H2-trophic methanogens in upper peat layer in a boreal northern peatland. Is there any evidence for southern peatlands too? This may partly explain relatively depleted d13C values of CH4 in the rhizosphere zone.

*We agree with the reviewer that elevated levels of H2 in the upper rhizosphere below Astelia lawns may indicate production of CH4 by hydrogenotrophic methanogenesis. We will include this possible explanation. We kindly refer to our answer to general comment 4 and 5 in the "general comments" section for more details.*

- Comment 2, lines 17-18: This can be misleading: 2nd hypothesis specified processes based on isotopic values, whereas here authors refer more to the concentrations/fluxes thereby considering rather 1st hypothesis. The latter, however was not supported. Please, rephrase. Also regarding the 2nd hypothesis, "less affected" is not appropriate for the hypothesis. Please, check the respective comment above.

*Indeed, the interpretation in lines 17-18 needs a better link to the hypotheses. Will be rephrased.*

- Comment 3, lines 19-21: This is not clear: How could roots of A. pumila appear under pools? Was this observed during coring? If so, then the conceptual diagram should demonstrate that roots of A. pumila expand below pools. Check and correct accordingly.

*This will be clarified: It is possible, but we do not know whether the roots appear under pools. Therefore we did not include this in the conceptual figure. We propose that roots control CH4 dynamics below pools only by releasing O2 that is used to consume CH4 thereby maintaining concentration gradients. We will clarify this on page 15, lines 19-21. Please refer to the "general comments" section, comment 6 for further details.*

- Comment 4, line 22: Of what, CH4 or oxygen?

*Will be specified. Please refer to the "general comments" section, comment 6 for details.*

- Comment 5, lines 23-26: This statement is unclear: whereas gas diffusion along gradient is clear for me (from pools to lawns) water movement is not the same. Pools are local depressions, so water should flow from lawns into pools. If this flow is so low, then the gas diffusion in opposite direction can be stronger, but this means almost standing water. In case there is a lateral flow of water (what is very natural), then the gas flow can't be counter to it. Therefore, I could understand the inflow of oxygen from lawns into pools, but not CH4 from pools to lawns. The overall picture may change if there is a slope, but then lawns and pools have to be arranged accordingly. Pools will get matter of those lawns which are elevated and transfer it downwards to other lawns. If there is a slope on the site, then the conceptual figure should somehow reflect it. Check!

*This comment was already answered above in the "general comments" section, comment 6.*

- Comment 6, lines 28-30: What is meant, suppression of methanogenesis or CH4 oxidation? This is important in context of measured isotope values. Please, specify.

*We did not check know whether the roots appear under pools, but it would be reasonable. Therefore, here suppression of methanogenesis is meant here as an explanation for low CH4 concentrations in pools. This will be specified in the revised discussion. Please refer to the "general comments" section, comment 6 for further details.*

- Comment 1, lines 8-9: This is not fully clear: the limiting factor for hydrogenotrophic methanogenesis is typically H2 which is very reactive, since C source is CO2. H2 was sufficient, CO2 concentrations were also available, so why was then H2-reduction methanogenesis although dominating (depleted d13C-CH4) but not intensive? Maybe other anaerobic processes (sulfate reduction) outcompeted methanogenesis?

*The reviewer is right, we need to check the argumentation here. Despite H2 concentrations and DIC:CH4 ratios suggest methanogenic conditions, CH4 production seemed to be limited*

*below pools even at peaking H2 levels. Very negative d13C values suggest that methanogenesis was thermodynamically unfavorable. Instead, methanogenesis might have been outcompeted by other electron accepting processes, such as sulfate reduction as suggested by the reviewer. Peaking H2 concentrations indicate that fermentation processes were active, but suggest that methane was produced only very locally if all. We will provide a more detailed explanation in the revised discussion.*

- − Comment 1: The section 4.4. is rather long and at several places contains repetitive text (e.g. page 17, lines 15-17, 21-23, 27-28; the effect of A. pumila roots was very clear, no need to repeat many times). I recommend condensing text strongly.

*This comment was already answered above in the "general comments" section, comment 7.*

- − Comment 1, lines 4-5:
- − Comment 2, line 5: On the figure (f), platform 3 instead platform 2 is denoted. Check!

*The reviewer is right, we need to check this.*

- − Comment 1: It was discussed a lot about lateral flows, which how, are not reflected in this conceptual diagram. It is also not clear if the site has elevation/slope property. In such a case, please demonstrate the respective relationships.

*This comment was already answered above in the "general comments" section, comment 6.*

References

Berger, S., Praetzel, L. S. E., Goebel, M., Blodau, C., and Knorr, K. H.: Differential response of carbon cycling to long-term nutrient input and altered hydrological conditions in a continental Canadian peatland, Biogeosciences, 15, 885-903, 2018.
Mastepanov, M., Sigsgaard, C., Mastepanov, M., Strom, L., Tamstorf, M. P., Lund, M., and Christensen, T. R.: Revisiting factors controlling methane emissions from high-Arctic tundra, Biogeosciences, 10, 5139-5158, 2013.
McEwing, K. R., Fisher, J. P., and Zona, D.: Environmental and vegetation controls on the spatial variability of CH4 emission from wet-sedge and tussock tundra ecosystems in the Arctic, Plant Soil, 388, 37-52, 2015.

---

## Author Response (AR1)

Dear reviewers,

we appreciate your thoughtful and detailed reviews! Your constructive comments certainly helped to improve our manuscript and particularly the discussion. We took your comments (black) into consideration as described below (blue):

**Point-by-point response to reviewer 1:**

General comments

This is a carefully done study about the production, oxidation and emission of CH4 in Patagonian bog, the results are of considerable interest and the paper is well written. However, some points need clarifying and certain statements require further justification.

Specific comments

1. the authors should not ignore that acetogenesis might be important in anaerobic environments when H2 partial pressures are high and temperatures are low. Acetogens can outcompete methanogens at low temperature, as many acetogens seem to have a higher growth rate at low temperature than most methanogens (Kotsyurbenko et al., 1996, 2001). If acetogenesis process is active in the bog, the δ3C value of actetate in the porewater will be largely decreased because of the substantial fractionation during acetate production from CO2 and H2. And resultantly, the 13C value of CH4 will also be lower and resulted in larger apparent isotopic fractionation factor (ac) between CO2 and CH4. Therefore, it's difficult to determine the relative importance of acetoclastic versus hydrogenotrophic methanogenesis pathway without the 13C value of acetate in this study.

*We agree with the reviewer, that the d13C-CH4 signal was surprisingly low and therefore needs an explanation. Indeed, a strong effect of only methanotrophy should result in a less negative signature of d13C in CH4. We attempt to explain the low d13C-CH4 signal by the occurrence of microsite that create a mixed isotopic signal from methane production and consumption. As the reviewer correctly points out, the occurrence of acetogenesis is a reasonable explanation for the surprisingly negative d13C-CH4 signatures. Reviewer 2 suggests occurrence of hydrogenotrophic methanogenesis as a possible explanation. We agree that both pathways could explain the pattern in the d13C-CH4 signal, although we propose in accordance with the reviewer that hydrogenotrophic methanogenesis was probably relatively more important below pools while acetoclastic methanogenesis in combination with acetogenesis seemed to contribute more below Astelia lawns. But as we did not quantify other parameters such as labile organic matter from roots, acetate concentrations or its carbon isotopic signatures, we cannot clearly determine the relative importance of both pathways for our study, as the reviewer correctly states.*
*In the revised discussion, we emphasize that the depleted d13C-CH4 signal is an unexpected result (page 14, lines 6-8, and lines 34-2 on page 15). We explain possible sources for depleted 13C-CH4 within the rhizosphere and discuss the possibility of the occurrence of acetogenesis (page 14, lines 16-22; page 15, line 2). The difficulties to separate the isotopic effects arising from methanogenic pathways are now mentioned (page 14, line 18 and page 15, lines 3-5). We tried to balance the suggestions of both reviewers by discussing the arguments for both, hydrogenotrophic and acetoclastic methanogenesis and acetogenesis, carefully (page 15, lines 2-11).*

2. In the first page, line 26-28, it's stated that: "Below the rhizosphere. . .. . .CH4 was predominantly produced by hydrogenotrophic methanogenesis". In fact, data in Figure 4def showed that the hydrogenotrophic pathway had higher contribution to CH4 in the pool, while the acetoclastic pathway must play relatively more important role for the CH4 production below the rhizosphere of Astelia Lawn. This is consistent with the supply of labile organic carbon from

the root exudates of Astelia. To sum up, I think it's difficult to conclude that CH4 is mainly produced from the hydrogenotrophic pathway below the rhizosphere of Astelia.

*We agree with the reviewer that hydrogenotrophic methanogenesis was probably relatively more important below pools while acetoclastic methanogenesis seemed to contribute more below Astelia lawns. This seemed to have been even true below the rhizosphere of Astelia lawns, but is a result hard to explain from the data obtained in our study. As the reviewer correctly points out, labile organic matter from roots could be a possible explanation. But as we did not quantify other parameters such as labile organic matter from roots, acetate concentrations or its carbon isotopic signatures, we cannot clearly determine the relative importance of both pathways for our study, as the reviewer correctly states in the general comment 1.*
*In the revised abstract, we rephrased that sentence on page 1, line 26-28. We emphasize in the discussion that the depleted d13C-CH4 signal is an unexpected result (page 14, lines 6-8, and lines 34-2 on page 15). We explain possible sources for depleted 13C-CH4 and discuss the possibility of the occurrence of acetogenesis (page 14, lines 16-22; page 15, line 2). The difficulties to separate the isotopic effects arising from methanogenic pathways are now mentioned (page 14, line 18 and page 15, lines 3-5). We tried to balance the suggestions of both reviewers by discussing the arguments for both, hydrogenotrophic and acetoclastic methanogenesis and acetogenesis, carefully (page 14, lines 18-20; page 15, lines 2-11).*

3. It's stated that mean root lifetimes of A. pumila has been estimated to be ~3-4 years. So, whether the production and oxidation of CH4 will be strongly affected in case of the turnover of large amounts of roots?

*It is an interesting point that turnover i.e. presence and activity of roots should largely determine the occurrence of microsites. One could expect that temporal and spatial expansion of microsites is not static but varies with root life time and turnover. However, as reviewer 2 pointed to a partly speculative discussion about microsites in the former version of the manuscript, we only briefly included this aspect into the revised discussion (page 14, lines 13-14).*

4. Please check Table 3, the data in the last three columns are in wrong places.
*Yes, the reviewer was correct. We corrected Table 3 accordingly. The data presented in this table are part of a larger dataset that is still analyzed at the moment. During this analyses we decided to re-analyze some peat samples to improve the data quality. Thus, some values slightly changed and the data in the table were updated.*

**Point-by-point response to reviewer 2:**

General comments

The MS of Münchberger and co-authors contributes to the knowledge on processes of methane turnover (production+oxidation) and transport in rarely studied southern bogs. Authors combined field sampling with the advanced analytics (porewater chemistry and stable isotope analyses) to report relationships and peculiar mechanisms between microrelief forms, dominating vegetation communities and the net processes affecting the CH4 efflux from the Patagonian peatland during two consecutive summer seasons. Field-based studies are critically important for understanding processes related to functioning of ecosystems and therefore interesting for the broad scientific community. Accepting the field experiments typically operate with much larger spatial and temporal variability in measured parameters (and as the result, relatively lower statistical power as compared to controlled conditions), still there are several issues which I would like to point out for the discussion and improvement. Below authors find general comments while specific recommendations and technical corrections are incorporated directly in the draft file attached.

1. First of all, the MS is rather long and too repetitive and descriptive. Thus, the Introduction is definitely too extended, especially regarding the common knowledge about methane in the very beginning and peatlands in general. Authors could immediately start the story of the importance of southern peatlands and have the necessary information on peatlands' biochemistry and vegetation specialty in there. Then the information on the isotope issue would be sufficient to formulate hypotheses without any loss of logic.

*We understand your concern and appreciate your suggestions of how the topic of the manuscript could be introduced in a more straightforward way. However, the strength of our dataset is the combination of both, a comprehensive chamber measurement campaign and advanced pore water chemistry. To address our manuscript to the readers of both communities, researcher focusing on gas exchange in peatlands as well as those mainly dealing with biogeochemical processes in the peat, we decided to shortly introduce the main mechanisms of both research areas. Southern peatlands are introduced only towards the end of the introduction since the rhizosphere effects on methane dynamics due to aerenchymatic vascular plants are of relevance for other peatlands with a dense cover of vascular plants such as rushes and sedges. We believe that starting the introduction with the importance of southern peatlands would result in a too narrowed topic of the manuscript. Therefore, and as reviewer 1 did not comment on the structure and length of the introduction, we hope that reviewer 2 could agree with our positions here.*
*Nevertheless, we agree that in parts the introduction could be substantially shortened. Descriptive parts as well as passages dealing with common knowledge about methane in the introduction are rephrased now. We thereby reduced the number of words in the introduction by about 200 words. Please see*
- *page 2: line 4, line 7, lines 10-18, line 24*
- *page 3: lines 7-9, lines 28-4 on page 4*

*As we also agree that parts of the discussion were too repetitive and descriptive, we shortened the discussion by another 100 words despite the interpretation of the d13C-CH4 signal concerning methanogenic pathways required more explanation as suggested by both reviewers. We kindly refer to our answer to general comment 4 to 7 below for details on this aspect.*

2. In the proposed hypotheses, it has to be clearer why pools are so much different from lawns in terms of methanogenesis pathways. This was not strait forward from the introduction; I suggest to omit statements as "remains less affected" because they are more confusing then explanatory; please, rephrase.

*We added some more information, in particular that pools are only scarcely vegetated (page 3, line 12). Hypotheses II was rephrased to "pore water CH4 and DIC concentration profiles and its carbon isotopic signatures below densely-rooted Astelia lawns reflect a distinct CH4 oxidation effect contrary to sparsely- or even non-rooted peat below pools" (page 2, lines 8-9) and in hypotheses III we changed "non-rooted" to "sparsely- or even non-rooted" (page 4, lines 8-9 and lines 11-12). Furthermore, "non-rooted" was changed to "sparsely- or even non-rooted" throughout the whole text.*

3. In the Methods section, I was confused with relatively short time (3 min) of chamber exposition even under the conditions of rather low atmospheric temperatures and low fluxes expected. Why also transparent and not opaque chambers were used for CH4 fluxes measurement?

*Indeed, chamber measurements with a chamber not connected to a fast gas analyzer need up to 30 minutes or more to determine methane fluxes. The chamber used in our study was connected to a portable gas analyzer (Ultraportable Greenhouse Gas Analyzer, 915-001, Los Gatos Research) with a 1 hz sampling rate. The instrument accuracy according to the manufacturer was < 2 ppb. Therefore, this instrument provides the opportunity to determine concentration changes even at low CH4 concentrations and within a short period of time (see for example McEwin et al., 2015; Berger et al., 2018 or Mastepanov et al., 2013 who used a similar gas analyzer with 1 hz sampling rate). Test measurements with a prolonged closure time and instantaneous on-site monitoring of gas concentrations changes within the chamber proved that a short measurement time was sufficient to determine the CH4 fluxes also at our study site. From this, we could furthermore exclude that zero fluxes are a methodological artifact.*
*And indeed, despite the short measurement time and low fluxes, only 105 of 537 flux measurements showed a concentration increase with a slope not significantly different from zero (page 7, lines 12-13). During all other measurements (at least a small but) significant concentration change was observed already during the short measurement time down to absolute values of 0.01 mmol CH4 m-2 d-1 (page 9, lines 27-28).*
*We are also confident that low or zero fluxes were not an artifact despite the short measurement time and at low temperatures. Otherwise collars with low or zero fluxes would have shown a response to temperature. We kindly refer to figure S02 in the supplement provided to the manuscript. This figure show that most individual collars did not show any response to temperature. Please compare also to the explanation on page 7, lines 16-20.*
*We used transparent and opaque chambers since the Los Gatos gas analyzer can measure CH4 and CO2 simultaneously. Thus, our measurement campaign was designed to determine also the NEE. As the CH4 fluxes did not differ systematically between light and dark measurements, we included also the light measurements in our data set to increase the sample size for CH4 fluxes.*

4. Discussion section contains repetitive and partly speculative information and therefore is currently too long. For instance, in the discussion of results on 13C-CH4 depth profile (page 14, lines 5-8) authors seemingly "oversell" their results: "scattered between" may also indicate no significant difference (this is not clear from the data). Indeed, Fig. 4d demonstrates rather narrow d13C-CH4 range along the whole depth profile. So, in fact, d13C-CH4 signal alone was not informative enough to approve the strong oxidative properties of rhizosphere of A. pumila. I agree that both methanogenesis and oxidation may co-exist in close vicinity, but still it may not explain lack of d13C-CH4 variation between upper and lower horizons unless CH4 produced in the rhizosphere region is even more depleted in 13C than in deeper layers. The explanation of this phenomenon because of "more reduced...microsites" is not fully clear. More than below the rhizosphere? Why?

*We understand that parts of the discussion are too repetitive and partly speculative. The comparatively small shift of the mean d13C-CH4 signature to more enriched values within the rhizosphere was a surprising and unexpected result also to us, so we intended to discuss*

*this more thoroughly – probably going too far. Indeed, a strong effect of only methanotrophy should result in a less negative signature of d13C in CH4 compared to the signature below the rhizosphere. Therefore we completely agree with the reviewer, that the d13C-CH4 signal alone does not provide a clear indication for oxidative effects in the rhizosphere. Taking into account the near-zero CH4 emissions, high DIC:CH4 ratios and a d13C-CH4 depth pattern not following the d13C-CO2 and a depleted d13C-CO2 signal, we can nevertheless only explain these results by a strong effect of methanotrophy.*

*In addition, we attempted to explain the lack of variation in the mean d13C-CH4 signal. Throughout the rhizosphere, the d13C-CH4 signal was associated with a wider standard variation of replicate samples compared to deeper peat layers. Our possible explanation for this is that the mean d13C-CH4 signal represents a mixed signal from methane production and consumption and, thus, indicates a co-existence of locally distinct aerobic and anaerobic microsites. Maybe our explanations were too detailed here and thus became speculative. In fact, the word scatter was indeed used incorrectly here to describe the wider standard variation.*

*As the reviewer correctly points out, the occurrence of hydrogenotrophic methanogenesis is a possible explanation for surprisingly negative d13C-CH4 signatures. Reviewer 1 suggests occurrence of acetogenesis as a further possible explanation. We agree that both pathways could explain the pattern in the d13C-CH4 signal, although we propose in accordance with reviewer 1 that hydrogenotrophic methanogenesis was probably relatively more important below pools while acetoclastic methanogenesis in combination with acetogenesis seemed to contribute more below Astelia lawns. But as we did not quantify other parameters such as labile organic matter from roots, acetate concentrations or its carbon isotopic signatures, we cannot clearly determine the relative importance of both pathways for our study. We kindly refer to our answer to general comment 5 below for more details.*

*In the revised discussion, we replaced "scatter"/"scattering" by "range"/"ranging"/"variation" (page 11, line 27; page 14, line 5 and 11). By merging paragraphs in section 4.2 (please see the "specific comments section, comment 3 on page 14), repetitive and speculative information in the discussion were reduced. We mention that the lack of d13C-CH4 variation between upper and lower horizons resulting in a comparatively negative δ13C-CH4 signature throughout the rhizosphere is an unexpected result (page 14, lines 6-8, and lines 34-2 on page 15). We discuss now which processes may have resulted in such 13C-CH4 signatures (page 14, lines 16-22) and tried to balance the suggestions of both reviewers by discussing the arguments for both, hydrogenotrophic and acetoclastic methanogenesis and acetogenesis, carefully (page 15, lines 2-9). The misleading statement "more reduced... microsites" was removed.*

5. Contribution of acetoclastic pathway to methanogenesis in the rhizosphere of A. pumila was not convincingly verified (e.g. page 14, lines 23-27) and looks therefore speculative: having acetoclastic methanogenesis and co-existence of oxidation should generate much more enriched d13C-CH4 values in comparison to deep peat. Fig. 4d cannot support this. Seemingly, change of fractionation factor with depth was not significant either. The available data are not enough to approve existence of acetoclastic methanogenesis, and this has to be acknowledged.

*We agree with the reviewer, that a clear effect of methanotrophy and acetoclastic methanogenesis on the d13C-CH4 signature should result in an enriched d13C-CH4 signal and a distinctly lower fractionation factor throughout the rhizosphere compared to deeper peat layers. We were thus surprised by the small (and probably not statistically significant) change in the fractionation factor. Thus, we attempt to explain a fractionation factor in the overlap range of ac from hydrogenotrophic and acetoclastic methanogenesis.*

*As already observed for the d13C-CH4 signal, also the fractionation factor shows a wider standard variation of replicate samples within the rhizosphere. The standard variation of the fractionation factor even tended to be larger with increasing depth down to the lower boundary of the rhizosphere. This pattern comes along with a presumably with depth decreasing root density (Fritz et al., 2011) in these depths. We therefore interpret this as a further indication for the occurrence of microsites as lower root density makes a more*

*heterogeneous peat matrix more likely and thus a higher variation in the fractionation factor between sampling sites.*

*We agree with the specific recommendation on page 15, comment 1 that occurrence of hydrogenotrophic methanogenesis at elevated H2 levels in surface peat layers might be a possible source for depleted 13C and an explanation for only small changes and a comparatively higher standard variation in the fractionation factor. Another reasonable explanation is the occurrence of acetogenesis, as suggested by reviewer 1.*

*We underline in the revised discussion that only small changes in the fractionation factor between upper and lower horizons are an unexpected result if methanotrophic effects are assumed (page 14, line 4). We explain possible sources for depleted 13C-CH4 within the rhizosphere and discuss the possibility of the occurrence of acetogenesis now (page 15, lines 2). The difficulties to separate the isotopic effects arising from methanogenic pathways are mentioned (page 14, line 18 and page 15, lines 3-5). As we did not quantify other parameters such as labile organic matter from roots, acetate concentrations or its carbon isotopic signatures, the possibility of both, hydrogenotrophic and acetoclastic methanogenesis is discussed (page 15, lines 5-11).*

6. Another critical point is again a speculative discussion of the results on pools and lateral flows on the site (page 15, lines 23-35). Explanations on gas diffusion along gradient were clear for me (from pools to lawns) but water movement is not the same. Pools are local depressions, so water should flow from lawns into pools. If this flow is so low, then the gas diffusion in opposite direction can be stronger, but this means almost standing water. In case there is a lateral flow of water (what is very natural), then the gas flow can't be counter to it. Therefore, I could understand the inflow of oxygen from lawns into pools, but not CH4 from pools to lawns. The overall picture may change if there is a slope, but then lawns and pools have to be arranged accordingly. Pools will get matter of those lawns which are exposed higher and transfer it downwards to other lawns. If there is a slope on the site, then the conceptual figure should somehow reflect it. Such important information was not provided in Mat&Meth or any other parts of MS.

*We thank the reviewer for this careful and critical examination of our concept. We attempted to explain low CH4 concentrations in the pore water below pools, as the isotopic signals below pools did not indicate a methanotrophic effect. Upward diffusion of CH4 against the concentration gradient is not possible and CH4 emissions were low, so we can only explain this by lateral exchange of CH4.*

*Indeed, pools are local depressions in the micro-relief, and we therefore agree that water should flow from lawns into pools. Nevertheless, the micro-relief is not very pronounced at our study site with Astelia lawns being elevated by only about 5-20 cm above the water table and the pool surface. This is indicated in the conceptual figure. We therefore assume, that the micro-relief does not exert much impact on the water flow in deeper peat layers and water flow from lawns to pools should be restricted to the uppermost decimeters of the peat profile. In contrast, the rhizosphere stretches over almost 2 m within highly decomposed peat. So we propose that there is a large zone with negligible water flow throughout the rhizosphere. Due to low water movement, diffusive transport dominates and both, CH4 transport from pools to lawns and O2 transport from lawns to pools could be reasonable.*

*Some information about the flat micro-relief is now included in the site description (page 4, line 27). We specify in the revised discussion that CH4 may be transported by diffusion and that the micro-relief is not pronounced (page 15, lines 29-32). Furthermore, repetitive explanations in section 4.3 were shortened by about 60 words.*

7. The section 4.4. is rather long and at several places contains repetitive text (e.g. page 17, lines 15-17, 21-23, 27-28; the effect of A. pumila roots was very clear, no need to repeat many times). I recommend condensing text strongly.

*We understand that the interpretation of the observed emission pattern is repetitive concerning biogeochemical processes in the peat. We carefully shortened repetitive explanations in paragraph 4.4., starting in line 12 on page 17 by 150 words.*

8. Depending on the available information from authors, the conceptual Fig. 6 can be changed (see more detailed comments in the text).

*We kindly refer to our answer to general comment 6.*

Specific recommendations and technical corrections incorporated in the draft file of the manuscript

– Comment 1, line 25: How was this tested? If so, how far from root surface the "suppressive effect" is possible?
*This was not tested and, thus, this conclusion was removed here.*
– Comment 2, lines 30-32: Please, rephrase the sentence in a more simplistic way. Too difficult to read and understand.
*Was rephrased.*

– Comment 1, lines 6-7: Please, check whether this value is still relevant. There is no recent reference cited.
*References were checked, but no recent reference was found splitting CH4 emissions from wetlands into sources from different types of wetlands explicitly mentioning peatlands. Instead, even very recent publications like Rinne et al. (2018) still cite Aselmann and Crutzen (1989) as a reference for CH4 emissions from a specific type of wetland. Furthermore and for example, Dlugokencky et al. (2011) cite Bousquet et al. (2006) who give an estimate of 30% of wetlands contributing to global CH4 emissions (in agreement with Kirschke et al, 2013) of which bogs and tundra make up 9% and swamps 21%. To conclude, we believe the estimate given here is still reasonable and there is – to the best of our knowledge - no recent reference for the contribution of peatlands to global annual CH4 emissions.*
– Comment 2, line 7: 28-fold
*This technical correction was changed accordingly.*
– Comment 3, lines 10-12: What about low temperatures? I am not sure bogs do exist in tropics (excluding mountain regions).
*Some information about the importance of temperature was added.*
– Comment 4, line 17: This is not clear: is this prerequisite for the oxidation? Not trapped=not oxidized? If so, how exactly could CH4 be trapped and how does CH4 oxidation occur in e.g. rice paddies?
*The sentence was rephrased.*
– Comment 5, line 30: (delta 13C)
*Was added.*
– Comment 6, line 32: Actually, values could go even to positive range.
*The range was changed to "-25 to +10‰".*

– Comment 1, line 17: Not clear. Do authors mean, reflect the signal of methanogenesis type? Please, rephrase.
*The hypothesis was changed. Please refer to the "general comments" section, comment 2.*
– Comment 2, lines 19-20: How is this known? From the introduction above, it is not clear the pools are without (vasular) vegetation. Northern pools often contain aerenchimatous plants of different species compared to lawns and hummocks, or at least Sphagnum. Please, clarify above.
*This comment was already answered above in the "general comments" section, comment 2.*

- Comment 3, line 23: I find the Introduction a bit too extended, especially regarding the common knowledge about methane in the very beginning and peatlands in general. Authors could immediately start the story of the importance of southern peatlands and have the necessary information on peatlands' biochemistry and vegetation specialty in there. Then the information on the isotope issue would be sufficient to formulate hypotheses without any loss of logic.

*This comment was already answered above in the "general comments" section, comment 1.*

- Comment 1, line 4: Not any more? Should it be in Present Tense as the previous sentence?

*The tense was corrected in the description of the study site.*
- Comment 2, line 5: Does this mean pools were nevertheless vegetated? If so, which species dominated? What was the bottom of such pools?

*The vegetation in the pools is described in more detail now.*
- Comment 3, line 6:

*The tense was corrected in the description of the study site.*
- Comment 4, line 7: Liter of what, peat? For peat, could you provide other volumetric dimensions, e.g. dm-3?

*The dry weight is given as liter of peat soil and this information was added. The volumetric dimension was not changed in order to stick to the units given in the cited reference.*

- Comment 1, line 16: This could be too short time for CH4 flux measurement especially if outside temperatures were relatively low. How was it determined? Could zero fluxes be the reason of short exposure time?

*This comment was already answered above in the "general comments" section, comment 3.*

- Comment 1, line 4: was

*"Was" was added and the sentence split into two parts to improve readability.*
- Comment 2, line 24: How? What was the volume of the sample? 3 ml? For such small volumes a separate device (Small volume unit) is necessary. Please, expalin.

*Yes, the reviewer is correct. The missing information in the description of the device and measurement routine was added (page 8, lines 19-20).*
- Comment 3, line 30: Confusing: organic or inorganic? Please correct.

*Was changed to "dissolved inorganic carbon".*

- Comment 1, line 22: Again, liter is not clear for peat as volume containing roots.

*"Liter of peat" was added.*
- Comment 2, line 23: Please, provide here a value with the reference to study (studies). This will help better compare the differences between plant species.

*This information was added.*

- Comment 1, line 1:

*Was changed accordingly.*
- Comment 2, line 1: This is confusing: zero flux is not detectable (otherwise it is a positive or negative). Please, rephrase. I am still wondering if outside air temperature is -0.5 C, how could 3 min be enough to measure any CH4 flux.

*We rephrased this phrase explaining that despite near-zero fluxes we were still able to detect these low magnitude fluxes. Nevertheless, below a certain range, fluxes were set to zero as they did not differ significantly from zero (page 9, lines 27-28). Concerning a short*

*measurement time at low temperature, we refer to our answer to comment 3 in the "general comments" section.*

  – Comment 1, lines 26-27: There was no rhizosphere below pools, so what then caused the gradient?

*The upward diffusion from deep peat below pools was caused by low pore water CH4 concentrations in upper peat layers below pools. Here, we changed "within the rhizosphere" to "throughout the rhizosphere of Astelia lawns and in corresponding depths below pools". We attempt to explain low pore water CH4 concentrations below pools by lateral transport of CH4 from pools to lawns (see section 4.3 in the discussion).*

  – Comment 2, line 28: Suggest to rephrase: Carbon isotopic values in pore water and apparent fractionation.

*The section title was rephrased.*

  – Comment 1, line 7: fractionation

*This technical correction was changed accordingly.*

  – Comment 2, line 22: with?

*We added "in" here.*

  – Comment 1, lines 30-31: Please, check the definitions: typically, "alternative" means alternative to oxygen. So, oxygen cannot be alternative to itself.

*This phrase was changed accordingly.*

  – Comment 1, lines 6-7: With this, authors attempt to oversell their results: "scattered between" seemingly indicate no significant difference. Indeed, Fig. 4d demonstrate rather narrow d13C-CH4 range along the whole depth profile. So, in fact, d13C-CH4 signal alone was not informative enough to approve the strong oxidative properties of rhizosphere of A. pumila. I agree that both methanogenesis and oxidation may co-exist in close vicinity, but still it may not explain lack of d13C-CH4 variation between upper and lower horizons unless CH4 produced in the rhizosphere region is even more depleted in 13C than in deeper layers. The explanation of this phenomenon because of "more reduced...microsites" is not fully clear. More than below the rhizosphere? Why?

*This comment was already answered above in the "general comments" section, comment 4.*

  – Comment 2, line 17: According to this oxidation concept, the most 13C enriched CH4 has to be allocated at the shallowest depth. However, in contrast, it is ca. 10‰ more depleted than next depth levels (20-50 cm). In addition, d13C-CO2 is relatively more enriched that in deeper layers. How is this possible?

*We inspected again our dataset to explain this. During a measurement, the isotopic signal of each sample is determined repetitively. So in fact, the signal determined from one sample is a mean of many measurements. To further improve the data quality, we excluded the isotopic signal of one sample with an elevated SD. This results now in a less enriched mean of d13C-CO2 and a slightly lower ac value in the uppermost peat layer. Figure 4e and 4f were corrected accordingly.*

*The sampling devices were installed below the water table, but only mean of water table is given in the figures. Thus, the line denoting the water table in the figures was not exactly at the correct position. This line was slightly shifted and figures 4 and 5 updated accordingly. So, in the uppermost depth not much influence from roots can be expected. Accordingly, the CH4 was not enriched due to methanotrophic effects, but comparatively depleted by methanogenesis (please compare to answers to comment 4 and 5 in the general comment section).*

- Comment 3, lines 17-22: This information is already repetition of the message above. I suggest to merge both parts telling the story as here but with the reference to results as in the previous paragraph. Otherwise, it is excessive.

*We followed the helpful suggestion of the reviewer and merged both paragraphs.*

- Comment 4, lines 23-26: This contradicts to the data measured: having acetoclastic methanogenesis and co-existence of oxidation should generate much more enriched d13C-CH4 values in comparison to deep peat. Fig. 4d cannot support this. Seemingly, change of fractionation factor with depth was not significant either. The available data are not enough to approve existence of acetoclastic methaogenesis. Please, discuss this

*This comment was already answered in the "general comments" section, comment 4 and 5.*

- Comment 5, lines 27-28: Again, there is not enough evidence to support the hypothesis. As it is stated, this is speculation and should be rephrased.

*We changed "supports" to "would support" and kindly refer to our answer to general comment 5 in the "general comments" section.*

- Comment 6, lines 29-30: Yes, but it was small below the rhizosphere too! Speculation!

*This comment was already answered in the "general comments" section, comment 4 and 5.*

- Comment 7, line 32: They also increased at the very top of profile. No information about significance of differences in fractionation factor between depths is provided.

*This comment was already answered in the "general comments" section, comment 5.*

- Comment 1, line 6: This may also mean occurrence of hydrogenotrophic methanogenesis in anaerobic rhizosphere zones. For example, Galand et al. (2002) FEMS, demonstrated dominance of H2-trophic methanogens in upper peat layer in a boreal northern peatland. Is there any evidence for southern peatlands too? This may partly explain relatively depleted d13C values of CH4 in the rhizosphere zone.

*We included this explanation and kindly refer to our answer to general comment 4 and 5 in the "general comments" section for more details.*

- Comment 2, lines 17-18: This can be misleading: 2nd hypothesis specified processes based on isotopic values, whereas here authors refer more to the concentrations/fluxes thereby considering rather 1st hypothesis. The latter, however was not supported. Please, rephrase. Also regarding the 2nd hypothesis, "less affected" is not appropriate for the hypothesis. Please, check the respective comment above.

*This was clarified and the discussion concerning pools better linked to the hypotheses.*

- Comment 3, lines 19-21: This is not clear: How could roots of A. pumila appear under pools? Was this observed during coring? If so, then the conceptual diagram should demonstrate that roots of A. pumila expand below pools. Check and correct accordingly.

*It is possible, but we do not know whether the roots appear under pools. Therefore we did not include this in the conceptual figure. We clarified that roots control CH4 dynamics below pools only by releasing O2 that is used to consume CH4 thereby maintaining concentration gradients. Please refer to the "general comments" section, comment 6 for further details.*

- Comment 4, line 22: Of what, CH4 or oxygen?

*Of CH4, this is specified now. Please refer to the "general comments" section, comment 6 for details.*

- Comment 5, lines 23-26: This statement is unclear: whereas gas diffusion along gradient is clear for me (from pools to lawns) water movement is not the same. Pools are local depressions, so water should flow from lawns into pools. If this flow is so low, then the gas diffusion in opposite direction can be stronger, but this means almost standing water. In case there is a lateral flow of water (what is very natural), then the gas flow can't be counter to it. Therefore, I could understand the inflow of oxygen from lawns into pools, but not CH4 from pools to lawns. The overall picture may change if there is a slope, but then lawns and pools have to be arranged accordingly. Pools will get matter of those lawns which are elevated and transfer it downwards to other lawns.

If there is a slope on the site, then the conceptual figure should somehow reflect it. Check!

*This comment was already answered above in the "general comments" section, comment 6.*

– Comment 6, lines 28-30: What is meant, suppression of methanogenesis or CH4 oxidation? This is important in context of measured isotope values. Please, specify.

*We did not check know whether the roots appear under pools, but it would be reasonable. Therefore, here suppression of methanogenesis is meant as an explanation for low CH4 concentrations in pools. This is specified in the revised discussion. Please refer to the "general comments" section, comment 6 for further details.*

– Comment 1, lines 8-9: This is not fully clear: the limiting factor for hydrogenotrophic methanogenesis is typically H2 which is very reactive, since C source is CO2. H2 was sufficient, CO2 concentrations were also available, so why was then H2-reduction methanogenesis although dominating (depleted d13C-CH4) but not intensive? Maybe other anaerobic processes (sulfate reduction) outcompeted methanogenesis?

*We changed the sentences stating that despite H2 concentrations and DIC:CH4 ratios suggest methanogenic conditions, CH4 production seemed to be limited below pools even at peaking H2 levels. Very negative d13C values suggest that methanogenesis was thermodynamically unfavorable. We included that methanogenesis might instead have been outcompeted by other electron accepting processes, such as sulfate reduction as suggested by the reviewer. Peaking H2 concentrations indicate that fermentation processes were active, but suggest that methane was produced only very locally if at all. Please refer to page 16, lines 10-16.*

– Comment 1: The section 4.4. is rather long and at several places contains repetitive text (e.g. page 17, lines 15-17, 21-23, 27-28; the effect of A. pumila roots was very clear, no need to repeat many times). I recommend condensing text strongly.

*This comment was already answered above in the "general comments" section, comment 7.*

– Comment 1, lines 4-5:
– Comment 2, line 5: On the figure (f), platform 3 instead platform 2 is denoted. Check!

*Was changed to platform a and b as actually in the field, we determined water table fluctuations at platform 1 and platform 3 constructed for chamber measurements.*

– Comment 1: It was discussed a lot about lateral flows, which how, are not reflected in this conceptual diagram. It is also not clear if the site has elevation/slope property. In such a case, please demonstrate the respective relationships.

*This comment was already answered above in the "general comments" section, comment 6.*

  - Reduced repetitive, descriptive and speculative information in paragraph 4.2, 4.3, 4.4 by altogether about 200 words.
- We corrected the data in Table 3.

[revised manuscript text omitted]